# Evidential Copula Concept Embedding Models

**Yanjie Qiu** [1]  **Xiaodong Yue** [2][†]  **Xuhui Fan** [3]  **Yufei Chen** [4]  **Jie Shi** [1]  **Wei Liu** [4]

## Abstract

Concept Embedding Models (CEMs) advance interpretable AI by extending Concept Bottleneck Models (CBMs) through semantic concept embeddings, providing an important solution in high-stakes domains such as medical diagnosis where accuracy and interpretability are critical. However, a fundamental limitation persists: existing CEMs inherently assume concept independence, critically overlooking the highly complex dependencies among concepts. To address this, we propose an Evidential Copula Concept Embedding Model (EC-CEM) that redefines the joint distribution over concepts, capturing inter-concept dependencies while maintaining a flexible structure that decouples the marginal concept distributions from their dependency structure. In particular, EC-CEM relaxes the concept independence assumption and uniquely integrates Copula theory with evidential deep learning to define a joint distribution over concepts. The proposed EC-CEM also develops two training objectives that aim at classification and concept modeling simultaneously. We provide theoretical justification via variational inference and demonstrate empirical superiority through extensive experiments.

## 1. Introduction

Interpretability in deep learning (Chakraborty et al., 2017; Zhang & Zhu, 2018; Linardatos et al., 2020; Saeed & Omlin, 2023; Mersha et al., 2024) has been critical for enhancing trust and accountability in high-stakes applications such as healthcare, finance, and autonomous systems. Existing efforts (Selvaraju et al., 2017; Lei et al., 2024; Lundberg &

[1]School of Computer Engineering and Science, Shanghai University, Shanghai, China [2]Institute of Artificial Intelligence, Shanghai University, Shanghai, China [3]School of Computing, Macquarie University, Sydney, Australia [4]School of Computer Science and Technology, Tongji University, Shanghai, China. Correspondence to: Xiaodong Yue <yswantfly@shu.edu.cn>.

*Proceedings of the 43rd International Conference on Machine Learning*, Seoul, South Korea. PMLR 306, 2026. Copyright 2026 by the author(s).

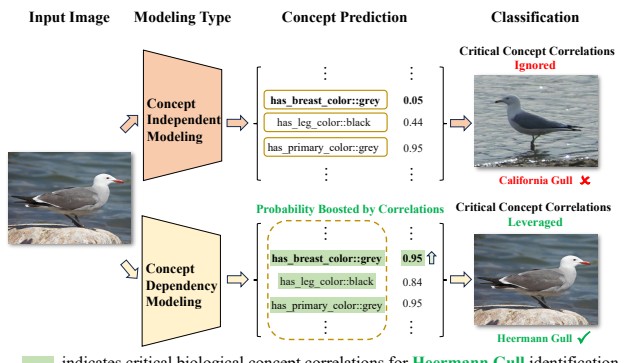

*Figure 1.* Comparison of independence modeling (top) versus dependency modeling (bottom) frameworks. The dependency approach captures critical information through inter-concept correlations.

Lee, 2017) usually provide interpretability for classification at the cost of effectiveness. Meanwhile, Concept Embedding Models (CEMs) (Espinosa Zarlenga et al., 2022; 2023) represent a significant step forward, extending Concept Bottleneck Models (CBMs) (Koh et al., 2020; Yuksekgonul et al.; Steinmann et al., 2024; Gao et al., 2025; Lin et al., 2026) to provide valuable concept-level explanations and semantic concept embeddings for model predictions.

However, the concept independence assumption in existing CEMs ignores critical correlations inherent in real-world data. As shown in Figure 1, distinguishing visually similar bird species relies on concept co-occurrence patterns—for instance, specific leg colors strongly imply specific plumage patterns. By treating concepts independently, CEMs fail to leverage these concept correlations, leading to misclassification when visual evidence is ambiguous.

Modeling these dependencies is non-trivial. Significant progress has been made by approaches like SCBM (Vandenhirtz et al., 2024), which captures correlations by modeling concept logits with a multivariate Gaussian distribution. While this approach effectively introduces dependency modeling, it faces a structural challenge inherent to the Gaussian parameterization: the representation of marginal uncertainty and inter-concept dependency is coupled within the shared covariance matrix, leading to an implicit trade-off where increasing uncertainty can inadvertently dilute the modeled correlation. To address this limitation, we propose leverag-

ing Copula theory, which provides a principled mechanism to decouple marginal distributions (uncertainty) from the dependency structure (correlation), making it ideal for modeling concept probabilities constrained to $[0, 1]$.

To realize this decoupling, we propose **E**vidential **C**opula **C**oncept **E**mbedding **M**odels (**EC-CEM**). Unlike previous methods that entangle uncertainty and correlation, EC-CEM adopts a divide-and-conquer strategy: it first employs Evidential Deep Learning (EDL) (Sensoy et al., 2018) to predict Beta-distributed parameters for each concept, explicitly capturing marginal epistemic uncertainty. Subsequently, a Gaussian Copula (Nelsen, 2006) is introduced to link these independent marginals into a coherent joint distribution. This construction ensures that the learning of inter-concept dependencies (via the Copula correlation matrix) does not distort the faithfulness of individual concept uncertainty estimates.

Within this probabilistic framework, we define a variational inference scheme that enables tractable optimization of both concept inference and downstream classification. Crucially, this joint optimization allows the model to learn task-aware correlations—prioritizing dependencies that are most discriminative for the target class.

In summary, our main contributions are as follows:

- **Decoupled Probabilistic Framework**: We propose EC-CEM, which leverages Copula theory to decouple marginal uncertainty from dependency structure, resolving the structural trade-off inherent in existing Gaussian-based methods.

- **Evidential Copula Construction**: We design a novel architecture that integrates Evidential Deep Learning with Gaussian Copulas. By employing Beta distributions as marginals, our approach provides a theoretically grounded solution for modeling bounded concept probabilities and their correlations.

- **Variational Joint Optimization**: We derive a variational inference objective that enables end-to-end optimization of concept inference and downstream classification. Experiments demonstrate that EC-CEM outperforms strong baselines, particularly yielding significant gains in sparse-concept scenarios.

## 2. Related Works

### 2.1. Interpretable Models

**Concept Bottleneck Models (CBMs)** (Koh et al., 2020) are among the first to explicitly incorporate human-understandable concepts into neural networks as intermediate variables to improve interpretability and controllability. Building on this idea, subsequent work has explored

learning concepts from semantically meaningful input features or weaker forms of supervision (Margeloiu et al., 2021). Autoregressive CBMs (AR-CBM) (Havasi et al., 2022) mitigate information leakage by modeling concept dependencies autoregressively, predicting each concept conditioned on the input and all preceding concepts in a fixed order. Stochastic Concept Bottleneck Models (SCBM) (Vandenhirtz et al., 2024) instead model concept dependencies through a learned multivariate normal distribution over concept logits. For binary concepts, this logit-space approach relies on a sigmoid transformation to obtain probabilities and therefore does not provide a direct parametric distribution over concept probabilities. Crucially, such latent Gaussian formulations parameterize marginal logit dispersion and inter-concept coupling within a shared covariance structure, which complicates independent interpretation and control of per-concept uncertainty and joint dependency.

Other CBM extensions incorporate unsupervised concepts (Sawada & Nakamura, 2022), post-hoc transferability (Yuksekgonul et al.), or label-free interpretability (Oikarinen et al.). Probabilistic approaches like ProbCBMs (Kim et al., 2023) further integrate uncertainty estimates into embeddings. Despite these advances, CBM-based models often suffer from performance degradation under concept sparsity and struggle to balance interpretability with task accuracy.

**Concept Embedding Models (CEMs).** Recognizing limitations in balancing task performance and interpretability, Concept Embedding Models (CEMs) (Espinosa Zarlenga et al., 2022) introduce a paradigm utilizing high-dimensional learnable vectors for concept representation. CEM learns dual embeddings for each concept: active embedding $(e_i^+)$ for presence and inactive embedding $(e_i^-)$ for absence. These are combined through learned probability $\hat{p}_i = \sigma(W_s[e_i^+, e_i^-]^T + b_s)$ to form the final representation:

$$e_i = \hat{p}_i e_i^+ + (1 - \hat{p}_i)e_i^-. \qquad (1)$$

Furthermore, CEM introduces RandInt regularization to enhance intervention capabilities by stochastically substituting predicted probabilities with ground truth $c_i$ during training:

$$e_i = \begin{cases} (c_i e_i^+ + (1 - c_i)e_i^-) & \text{with probability } p_{\text{int}} \\ (\hat{p}_i e_i^+ + (1 - \hat{p}_i)e_i^-) & \text{with probability } (1 - p_{\text{int}}) \end{cases} \qquad (2)$$

The parameter $p_{\text{int}}$ controls substitution frequency during training, while inference relies solely on predicted probabilities. This strategy creates an ensemble effect that enhances responsiveness to manual concept interventions at inference time.

Based on the original CEM, evi-CEM (Gao et al., 2024) employs subjective logic (Jsang, 2018) and evidential learning theory (Sensoy et al., 2018) to quantify concept uncertainty. Evidential learning theory has been widely adopted

across diverse applications, including multi-view learning and medical-assisted diagnosis (Liu et al., 2024; 2025a;b;c). However, existing CEMs typically assume that concepts are mutually independent, a critical limitation that causes them to overlook the valuable information contained in inter-concept correlations.

## 2.2. Copula theory

Copula function (Nelsen, 2006) has become one of the most important tools to fully study the dependence structures between random variables. A $K$-dimensional Copula function $C(\cdot, \ldots, \cdot)$ is defined as a distribution over the unit cube $[0,1]^K$, where the marginal distribution of each variate is a Uniform distribution over the interval $[0,1]$. The notable Sklar's theorem (Sklar, 1959) states that when given a Copula function $C(\cdot, \ldots, \cdot) : [0,1]^K \rightarrow [0,1]$ and when given $K$ arbitrary marginal cumulative distribution functions (c.d.f.) $\{F_i(x_i)\}_{i=1}^K$, a multivariate c.d.f. $F(x_1, \ldots, x_K)$ with marginals $\{F_i(x_i)\}_{i=1}^K$ can be defined through the Copula function as:

$$P(x_1, \ldots, x_K) = C(F_1(x_1), \ldots, F_K(x_K)). \quad (3)$$

Furthermore, the joint probability density function (p.d.f.) $p(x_1, \ldots, x_K)$ can be obtained in terms of the Copula density function $c(F_1, \ldots, F_K) = \frac{\partial C(F_1, \ldots, F_K)}{\partial F_1 \cdots \partial F_K}$ and the marginal p.d.f. $f_i(x_i) = \frac{\partial F_i(x_i)}{\partial x_i}$ as:

$$p(x_1, \ldots, x_K) = c(F_1(x_1), \ldots, F_K(x_K)) \prod_{i=1}^K f_i(x_i). \quad (4)$$

**Gaussian Copula** There have been multiple choices of Copula functions, such as Archimedean Copula (Zhang et al., 2024), vine Copula (Czado & Nagler, 2022; Xu & Cao, 2023), and Gaussian Copula (Joe, 2014; Suh & Choi, 2016). The proposed EC-CEM framework employs the Gaussian Copula function, a model well-regarded in both statistics (Chen & Gutmann, 2019) and machine learning (Wang & Wang, 2019) as a light-weight yet expressive tool for dependence modeling. We specifically selected it for this work due to its mathematically tractable formulation and ease of correlation estimation. In detail, a Gaussian Copula function is parameterized by a correlation matrix $\mathbf{R} \in [-1,1]^{K \times K}$. Its p.d.f. can be written as:

$$c_{\mathbf{R}}(u_1, \cdots, u_K) = \frac{\phi_{\mathbf{R}}(\Phi^{-1}(u_1), \ldots, \Phi^{-1}(u_K))}{\prod_{k=1}^K \phi(\Phi^{-1}(u_k))}, \quad (5)$$

where $\phi_{\mathbf{R}}(\cdot, \ldots, \cdot)$ is the p.d.f. of multivariate Gaussian distribution with zero means and correlation matrix $\mathbf{R}$, $\Phi^{-1}(\cdot)$ is the inverse c.d.f. of a standard Gaussian distribution, and $\phi(\cdot)$ is the p.d.f. of the standard Gaussian distribution. A

visual illustration of Gaussian Copula can be seen in Figure 3, which involves a joint distribution of two Beta variables (Beta(5,2), Beta(2,5)) with correlation $\rho = 0.7$. The transformation modify dependence structure, with the Beta distribution properties remaining unchanged and closely matching theoretical curves.

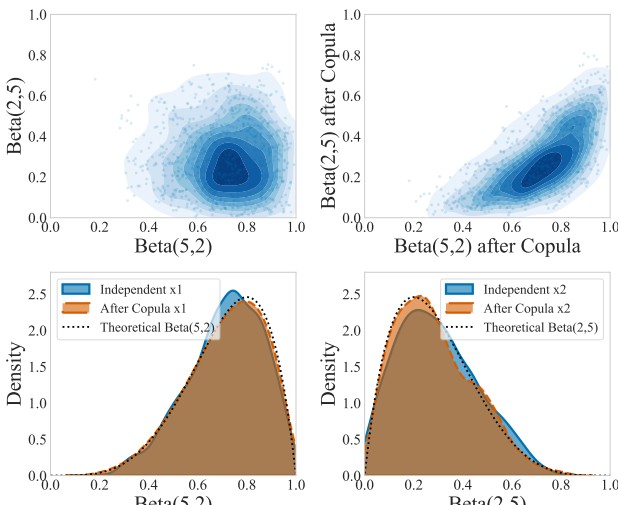

*Figure 3.* Gaussian Copula transformation applied to Beta(5,2) and Beta(2,5) distributions with correlation $\rho = 0.7$, showing preserved marginal distributions and modified dependence structure.

## 3. Method

This section presents the main methodology of the proposed EC-CEM model. From the model architecture perspective, we successfully integrate Gaussian Copula function to fully model dependencies between concept distributions. From the model inference perspective, we develop an efficient variational Bayesian method to learn the posterior distribution of concept existence probabilities.

Figure 2 visualizes the four stages in EC-CEM's architecture: Concept Embedding, Evidential Learning for concepts, Gaussian Copula Procedure, and the dual Training Objectives including classification and concept learning.

### 3.1. Concept Embedding

In the Concept Embedding stage, the input image $\mathbf{X}$ is first encoded into a latent embedding vector $\mathbf{h}$ through an image encoder $\Psi(\cdot)$, denoted as $\mathbf{h} = \Psi(\mathbf{X})$. Similar to CEM (Espinosa Zarlenga et al., 2022), this encoded feature $\mathbf{h}$ is then fed into two concept-specific layers $\varphi_k^+(\cdot), \varphi_k^-(\cdot)$ to obtain two concept-specific outputs $\mathbf{e}_k^+, \mathbf{e}_k^-$ as $\mathbf{e}_k^+ = \varphi_k^+(\mathbf{h}), \mathbf{e}_k^- = \varphi_k^-(\mathbf{h})$, where the signs of $\cdot^+$ and $\cdot^-$ denote the *presence* and *absence* of the $k$-th concept respectively.

This Concept Embedding stage encodes the image $\mathbf{X}$ into $K$ pairs of presence and absence embeddings $\{\mathbf{e}_k^+, \mathbf{e}_k^-\}_{k=1}^K$.

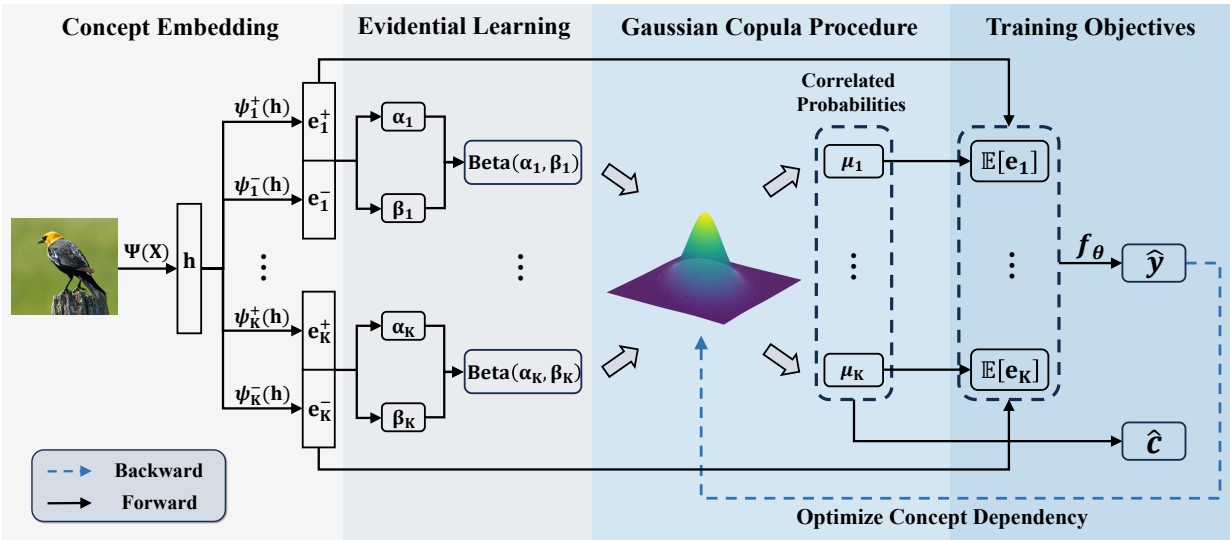

*Figure 2.* Framework of EC-CEM. Each concept is modeled using a Beta distribution derived from its concept embeddings. A Gaussian Copula integrates dependencies among concepts (illustrated here with bivariate Gaussian; the actual model uses K-dimensional Gaussian). Sampled values from Beta distributions are then adjusted according to copula-derived dependencies while preserving the original Beta parameters.

These pairs will be used to form the parameters of concepts' distributions (Gao et al., 2024) and also to the classification objective.

### 3.2. Evidential learning for concepts

Let $\mathbf{c} = \{c_1, c_2, ..., c_K\}$ denote $K$ binary concepts, with $c_k$ denoting the $k$-th concept. The subjective logic theory (Jsang, 2018) suggests that the existence probability of the $k$-th concept can be modeled through a Beta distribution as $\mu_k \sim \text{Beta}(\alpha_k, \beta_k)$, where $\alpha_k$ and $\beta_k$ encode the positive and negative evidence for the concept $c_k$. Using the concept embedding obtained in the above section, these positive and negative evidence can be formulated as (Gao et al., 2024):

$$\alpha_k = \text{ReLU}(\mathbf{W}_\alpha[\mathbf{e}_k^+, \mathbf{e}_k^-]^T + \mathbf{b}_\alpha) + 1, \quad (6)$$

$$\beta_k = \text{ReLU}(\mathbf{W}_\beta[\mathbf{e}_k^+, \mathbf{e}_k^-]^T + \mathbf{b}_\beta) + 1, \quad (7)$$

where $\mathbf{W}_\alpha$, $\mathbf{W}_\beta$ and $\mathbf{b}_\alpha$, $\mathbf{b}_\beta$ are learnable weights and biases of the linear layers.

### 3.3. Gaussian Copula Procedure

After obtaining the distribution over each individual concept's existence probability $\mu_k$, a Gaussian Copula function is introduced to model the joint distribution over all the $K$ concepts' existence probabilities. Their joint p.d.f. can be written as:

$$P(\mu_1, \ldots, \mu_K | \{\alpha_k, \beta_k\}_k, \mathbf{R})$$
$$= c_{\mathbf{R}}(F(\mu_1), \ldots, F(\mu_K)) \prod_{k=1}^{K} p(\mu_k; \alpha_k, \beta_k), \quad (8)$$

where $c_{\mathbf{R}}(\cdot, \ldots, \cdot)$ is the density function of Gaussian Copula function, with $\mathbf{R}$ as the correlation matrix.

Using Copula functions ensures that the marginal distribution over each existence probability $\mu_k$ from the joint distribution $P(\mu_1, \ldots, \mu_K | \{\alpha_k, \beta_k\}_k, \mathbf{R})$ still follows the same Beta distribution as $\mu_k \sim \text{Beta}(\alpha_k, \beta_k)$. At the same time, the correlation matrix $\mathbf{R}$ may well describe the dependencies between concepts. From a computational standpoint, the copula component scales as $O(K^2)$ in the number of concepts $K$, matching SCBM's multivariate Gaussian. For the concept sizes in our benchmarks, this introduces only a modest overhead relative to the backbone encoder.

### 3.4. Training objectives

EC-CEM develops two training objectives to facilitate learning of concept embedding and concept correlations: classification and concept fitting.

**Objective of classification** We predict the label $y \in \mathcal{Y}$ by using the *expected* concept embedding $\mathbb{E}[\mathbf{e}_k]$ with respect to $\mu_k$ as:

$$\mathbb{E}[\mathbf{e}_k] = \mu_k \mathbf{e}_k^+ + (1 - \mu_k)\mathbf{e}_k^-. \quad (9)$$

The final classification prediction $\hat{y}$ is then obtained by applying a classifier $\mathbf{f}_\theta(\cdot)$ with all the expected concept embeddings as inputs:

$$\hat{y} = \mathbf{f}_\theta(\mathbb{E}[\mathbf{e}_1], \mathbb{E}[\mathbf{e}_2], \ldots, \mathbb{E}[\mathbf{e}_K]). \quad (10)$$

**Objective of concept fitting** These observed concept $\{c_k\}_k$ can be used for the second objective of concept fitting, with

the $k$-th concept generated as:

$$c_k|\mu_k \sim \text{Bernoulli}(c_k; \mu_k), \forall k = 1, \ldots, K. \quad (11)$$

It is noted this objective does not require observing the whole set of concepts. A partial set of observed concepts is also workable since these concepts are independently modeled.

**Roles of different objectives** These two objectives play different roles in training. On one hand, the objective of concept fitting does not involve Copula correlation $\mathbf{R}$ and instead focuses on the concept embedding and Beta distribution parameters individually, since each concept $c_k$ is independently observed. On the other hand, all parameters, including the concept embedding parameters, Beta distribution parameters, Copula correlation $\mathbf{R}$, and classifier parameters $\boldsymbol{\theta}$, will be optimized in the objective of classification.

### 3.5. Inference

We develop variational inference methods to train the model. Given the joint distribution over random variables $\boldsymbol{\mu}, \mathbf{c}, \mathbf{y}$:

$$P(\boldsymbol{\mu}, \mathbf{c}, \mathbf{y}|\mathbf{h}) = p(\boldsymbol{\mu})P(\mathbf{c}|\boldsymbol{\mu})P(\mathbf{y}|\boldsymbol{\mu}, \mathbf{e}), \quad (12)$$

For each concept's existence probability $\mu_k$, we specify its prior as a Beta distribution $p(\mu_k) = \text{Beta}(1, 1)$; for each concept $c_k$, its prior is specified as $p(c_k|\mu_k) = \text{Bernoulli}(c_k; \mu_k)$, and for the classification label $\mathbf{y}$, its prior is specified as $P(\mathbf{y}|\boldsymbol{\mu}, \mathbf{e}) = \text{Categorical}(\mathbf{y}; \mathbf{f}_{\boldsymbol{\theta}}(\mathbb{E}[\mathbf{e}_1], \mathbb{E}[\mathbf{e}_2], \ldots, \mathbb{E}[\mathbf{e}_K]))$.

**Variational distribution of $\boldsymbol{\mu}$** As specified in the above section, the variational distribution over the concept existence probability vector $\boldsymbol{\mu} = [\mu_1, \ldots, \mu_K]^\top$ is set as a joint distribution which composes of a Gaussian Copula function and $K$ marginal Beta distributions:

$$q(\boldsymbol{\mu}|\mathbf{h}) = c_{\mathbf{R}}(F_1(\mu_1; \alpha_1, \beta_1), \ldots, F_K(\mu_K; \alpha_K, \beta_K))$$
$$\cdot \prod_{k=1}^{K} f_k(\mu_k; \alpha_k, \beta_k), \quad (13)$$

where $F_k(\mu_k; \alpha_k, \beta_k)$ and $f_k(\mu_k; \alpha_k, \beta_k)$ denote the c.d.f. and p.d.f. of the $k$-th Beta distribution $\text{Beta}(\mu_k; \alpha_k, \beta_k)$.

**Evidence Lower Bound (ELBO)** The Evidence Lower Bound (ELBO) for variational inference can be written as:

$$\log p(\mathbf{c}, \mathbf{y}|\mathbf{x})$$
$$= \log \int_{\boldsymbol{\mu}} \frac{p(\boldsymbol{\mu})P(\mathbf{c}|\boldsymbol{\mu})P(\mathbf{y}|\boldsymbol{\mu}, \mathbf{e})q(\boldsymbol{\mu}|\mathbf{h})}{q(\boldsymbol{\mu}|\mathbf{h})}\mathrm{d}\boldsymbol{\mu}$$
$$\geq \mathbb{E}_{q(\boldsymbol{\mu}|\mathbf{h})}\left[\log \frac{p(\boldsymbol{\mu})P(\mathbf{c}|\boldsymbol{\mu})P(\mathbf{y}|\boldsymbol{\mu}, \mathbf{e})}{q(\boldsymbol{\mu}|\mathbf{h})}\right]$$
$$= -\text{KL}[q(\boldsymbol{\mu}|\mathbf{h}) \| p(\boldsymbol{\mu})] + \mathbb{E}_{q(\boldsymbol{\mu}|\mathbf{h})}[\log p(\mathbf{y}|\boldsymbol{\mu}, \mathbf{e})]$$
$$+ \sum_{k=1}^{K} \mathbb{E}_{\mu_k \sim \text{Beta}(\mu_k; \alpha_k, \beta_k)}[\log p(c_k|\mu_k)]. \quad (14)$$

**KL-divergence term** The Kullback-Leibler (KL) divergence term $\text{KL}[q(\boldsymbol{\mu}|\mathbf{h}) \| p(\boldsymbol{\mu})]$ measures the difference between the variational posterior distribution $q(\boldsymbol{\mu}|\mathbf{h})$ and the prior distribution $p(\boldsymbol{\mu})$, thereby functioning as a regularization mechanism.

We can decompose this KL-divergence term into two separate terms, corresponding to the divergence between Beta distributions and the Copula correlation matrix independently.

$$\text{KL}[q(\boldsymbol{\mu}|\mathbf{h}) \| p(\boldsymbol{\mu})] = \underbrace{\text{KL}[\mathcal{N}(\mathbf{0}, \mathbf{R})\|\mathcal{N}(\mathbf{0}, \mathbf{I})]}_{\text{Copula Dependency Regularization}}$$
$$+ \underbrace{\sum_{k=1}^{K}\text{KL}[\text{Beta}(\mu_k; \alpha_k, \beta_k)\|\text{Beta}(\mu_k; 1, 1)]}_{\text{Marginal Regularization}}. \quad (15)$$

The complete derivation of the ELBO and the decomposition of the KL divergence term are detailed in Appendix A.

**Expected classification likelihood** There is no closed-form expression for the expected classification likelihood $\mathbb{E}_{q(\boldsymbol{\mu}|\mathbf{h})}[\log p(\mathbf{y}|\boldsymbol{\mu}, \mathbf{e})]$ term due to the complex form of $\mathbf{f}_{\boldsymbol{\theta}}(\cdot, \ldots, \cdot)$. Instead, $\mathbb{E}_{q(\boldsymbol{\mu}|\mathbf{h})}[\log p(\mathbf{y}|\boldsymbol{\mu}, \mathbf{e})]$ can be estimated by using Monte Carlo methods with $S$ samples as:

$$\mathbb{E}_{q(\boldsymbol{\mu}|\mathbf{h})}[\log p(\mathbf{y}|\boldsymbol{\mu}, \mathbf{e})]$$
$$\approx \frac{1}{S}\sum_{i=1}^{N}\mathbf{1}_{y_i=k} \cdot \sum_{\boldsymbol{\mu}^s \sim q(\boldsymbol{\mu}|\mathbf{h})}\log f_{\boldsymbol{\theta},k}(\mathbb{E}[\mathbf{e}_1^s], \ldots, \mathbb{E}[\mathbf{e}_K^s]),$$
$$(16)$$

where $f_{\boldsymbol{\theta},k}(\mathbb{E}[\mathbf{e}_1^s], \ldots, \mathbb{E}[\mathbf{e}_K^s])$ denotes the predicted probability for sample $i$ using the $s$-th Monte Carlo sample $\boldsymbol{\mu}^s$ and Equation (9) shows $\mathbb{E}[\mathbf{e}_1^s]$ is a transformation of $\mu_k$.

Directly sampling from the joint distribution over concept existence probabilities $\boldsymbol{\mu}$ is difficult. We adopt reparameterization techniques detailed in Algorithm 1 (Wang & Wang, 2019) and Gaussian Copula sampling in Algorithm 2 to sample $\boldsymbol{\mu}$. This expectation formulation enables the model to learn task-aware Copula dependencies optimized for downstream classification (such as disease diagnosis). Through

---

**Algorithm 1** Parameterization of Correlation Matrix

---

**Input:** Input image $\mathbf{x} \in \mathbb{R}^{H \times W \times C}$
**Output:** Lower triangular matrix $\mathbf{L}$
1: $\mathbf{h} = \text{Encoder}(\mathbf{x})$
2: $\mathbf{w} = \text{SoftPlus}(\mathbf{W_1} \cdot \mathbf{h} + \mathbf{b_1}) + 1$
3: $\mathbf{a} = \text{Tanh}(\mathbf{W_2} \cdot \mathbf{h} + \mathbf{b_2})$
4: $\mathbf{\Sigma} = \mathbf{w} \cdot \mathbf{I} + \mathbf{a}\mathbf{a^T}$
5: **return** $\mathbf{L} = \text{Cholesky}(\text{Normalize}(\mathbf{\Sigma}))$

---

**Algorithm 2** Generating concept existence probabilities vector $\boldsymbol{\mu}$ from its joint distribution with Copula structure

---

**Input:** Parameters $\{\alpha_k, \beta_k\}_k$ of concept existence probability distributions; lower triangular matrix $\mathbf{L}$
**Output:** Concept existence probability vector $\boldsymbol{\mu}$
1: **for** $k = 1 : K$ **do**
2: $\quad \widetilde{\mu}_k \sim \text{Beta}(\alpha_k, \beta_k)$
3: $\quad \widetilde{u}_k = \text{Beta}_{\text{CDF}}(\widetilde{\mu}_k; \alpha_k, \beta_k)$
4: $\quad \widetilde{z}_k = \Phi^{-1}(\widetilde{u}_k)$
5: **end for**
6: $\mathbf{z} = \mathbf{L}[\widetilde{z}_1, \ldots, \widetilde{z}_K]^\top$
7: $\mathbf{u} = \Phi(\mathbf{z})$
8: $\mu_k = \text{Beta}_{\text{ICDF}}(u_k; \alpha_k, \beta_k)$
9: **return** $\boldsymbol{\mu} = [\mu_1, \ldots, \mu_K]^\top$

---

this concept dependency modeling, the approach can also achieve an enhanced semantic concept representation $\mathbf{e}$.

**Expected concept likelihood** Since each concept label $c_k$ using a Bernoulli distribution, denoted as $c_k \sim \text{Bernoulli}(c_k; \mu_k)$, the likelihood function for all concept labels can be expressed as $p(\mathbf{c}|\boldsymbol{\mu}) = \prod_{k=1}^K (\mu_k)^{c_k}(1 - \mu_k)^{1-c_k}$. As a result, the corresponding expectation term can be obtained as:

$$\mathbb{E}_{q(\boldsymbol{\mu}|\mathbf{h})}\left[\log p(\mathbf{c}|\boldsymbol{\mu})\right]$$
$$= \sum_{k=1}^K (1 - c_k)\psi(\beta_k) - \psi(\alpha_k + \beta_k) + c_k\psi(\alpha_k). \quad (17)$$

It is noted that the expected concept likelihood term is factorized into the sum of individual concept likelihood. Consequently, each concept's expected likelihood is calculated based on its marginal distribution only, and is independent of the Copula correlation matrix.

# 4. Experiment

## 4.1. Experiment Settings

**Datasets and Metrics:** We evaluate EC-CEM on three benchmark datasets: Animals with Attributes 2 (AwA2) (Xian et al., 2019), containing 37,322 images from 50 animal classes with 85 attributes; Caltech-UCSD Birds-200-2011 (CUB) (Wah et al., 2011), consisting of 11,788

images from 200 bird species with 112 attributes following (Espinosa Zarlenga et al., 2022); and Fitzpatrick17k (F17k) (Groh et al., 2021), where we use the SkinCon-annotated subset of 3,218 images with 22 selected clinical concepts for skin disease classification following (Gao et al., 2024). We report Accuracy, AUC, and F1-score for both concept prediction and downstream classification. To assess semantic faithfulness of concept representations, we additionally report Concept Alignment Score (CAS) (Espinosa Zarlenga et al., 2022). Finally, we evaluate concept-level calibration using Concept Expected Calibration Error (Concept ECE), measuring the discrepancy between predicted concept probabilities and empirical correctness.

**Baseline:** EC-CEM is compared against established interpretable architectures from two families: Concept Bottleneck Models (CBM-sigmoid, CBM-logits, AR-CBM, SCBM) and Concept Embedding Models (CEM, evi-CEM). CBM-sigmoid and CBM-logits use sigmoid activation and raw logits respectively for concept prediction. AR-CBM employs an autoregressive structure $p_\theta(c_k|x, c_{1:k-1})$ to capture concept dependencies. SCBM models concept logits with a multivariate normal distribution and approximates concept probabilities via Monte Carlo sampling: $p(c_k|x) \approx \frac{1}{M}\sum_{m=1}^M \sigma(\eta_k^{(m)})$, where $\eta^{(m)} \sim \mathcal{N}(\mu(x), \Sigma(x))$. A detailed comparison and summary of the differences between EC-CEM, AR-CBM, and SCBM can be found in Appendix B.

**Settings:** All methods used ResNet-34 (He et al., 2016) as the backbone encoder with identical classifier heads. We trained for 200 epochs (50 for SkinCon) with early stopping (patience=15) using AdamW optimizer with weight decay $10^{-2}$ and dataset-specific learning rates: $10^{-4}$ (CUB), $5 \times 10^{-5}$ (AwA2), and $5 \times 10^{-4}$ (SkinCon). We report mean and standard deviation of the results across runs. All models are implemented in PyTorch and trained on an NVIDIA A100 GPU with 40GB memory.

## 4.2. Experimental Results

As shown in Table 1, our proposed EC-CEM significantly outperforms most baselines across multiple datasets, achieving both state-of-the-art classification accuracy through its task-aware Copula dependencies. These results highlight the importance of modeling concept correlations and underscore its benefits for both interpretability and classification accuracy.

## 4.3. Ablation Study

Removing the Gaussian Copula reduces EC-CEM to evi-CEM. While this ablated variant exhibits a slight performance drop compared to CEM due to evidential regularization (Pandey & Yu, 2023), EC-CEM leverages concept

*Table 1.* Performance comparison with popular methods on three datasets based on the concept metrics and classification metrics.

| Dataset | Method | Concept Metric | | | Classification Metric | |
|---|---|---|---|---|---|---|
| | | ACC | AUC | F1 | ACC | F1 |
| AwA2 | CBM-sigmoid | 96.53±0.09 | 98.90±0.07 | 93.80±0.17 | 87.57±0.10 | 80.12±0.37 |
| | CBM-logits | 97.02±0.11 | 99.15±0.08 | 95.07±0.22 | 88.29±0.56 | 80.84±0.64 |
| | AR-CBM | 97.91±0.11 | 98.91±0.04 | 95.63±0.19 | 90.44±0.45 | 84.09±0.62 |
| | SCBM | 97.98±0.06 | 99.63±0.01 | 96.14±0.12 | 91.62±0.22 | 85.63±0.02 |
| | CEM | 97.79±0.23 | 99.57±0.10 | 96.42±0.35 | 90.66±0.73 | 84.18±1.18 |
| | evi-CEM w/o correlation | 97.35±0.47 | 99.49±0.13 | 95.61±0.85 | 88.87±1.33 | 81.92±2.09 |
| | EC-CEM w/ correlation | **98.09±0.84** | **99.65±0.07** | **96.75±0.31** | **93.04±0.22** | **87.78±0.32** |
| SkinCon | CBM-sigmoid | 84.55±2.59 | 75.61±2.70 | 58.70±1.71 | 76.39±1.68 | 50.34±2.04 |
| | CBM-logits | 91.13±0.10 | 76.35±0.91 | 59.79±0.85 | 77.79±1.13 | 58.56±1.39 |
| | AR-CBM | 91.05±0.29 | 79.18±0.95 | 62.37±0.35 | 75.10±0.07 | 31.09±0.87 |
| | SCBM | 90.89±1.15 | 79.50±1.58 | **64.70±0.20** | 75.51±0.48 | 55.80±1.57 |
| | CEM | 91.09±0.01 | 78.58±1.35 | 55.89±0.31 | 77.89±0.70 | 58.44±1.43 |
| | evi-CEM w/o correlation | 90.40±0.39 | 80.73±0.62 | 63.58±0.74 | 77.69±1.03 | 58.15±0.13 |
| | EC-CEM w/ correlation | **91.16±0.09** | **81.87±0.25** | 63.08±0.78 | **78.52±0.93** | **62.41±1.34** |
| CUB | CBM-sigmoid | 75.57±1.00 | 82.38±0.73 | 56.89±0.73 | 75.26±0.13 | 23.06±0.10 |
| | CBM-logits | 93.46±0.78 | 95.78±0.69 | 79.95±1.70 | 75.20±0.06 | 23.01±0.23 |
| | AR-CBM | 93.17±0.47 | 96.32±0.39 | 81.39±0.82 | 73.11±0.21 | 23.49±0.34 |
| | SCBM | 93.25±0.18 | 96.39±0.10 | 81.52±0.21 | 74.78±0.32 | 23.72±0.22 |
| | CEM | 93.43±0.58 | 95.90±0.50 | 80.38±0.75 | 75.48±0.22 | 23.19±0.44 |
| | evi-CEM w/o correlation | 93.34±1.56 | 96.29±0.98 | 81.76±1.33 | 74.48±2.22 | 23.13±0.95 |
| | EC-CEM w/ correlation | **93.77±0.46** | **97.28±0.33** | **82.20±0.42** | **76.19±0.15** | **24.50±0.11** |

correlations to compensate for this trade-off, effectively recovering the accuracy and achieving better performance over both baselines.

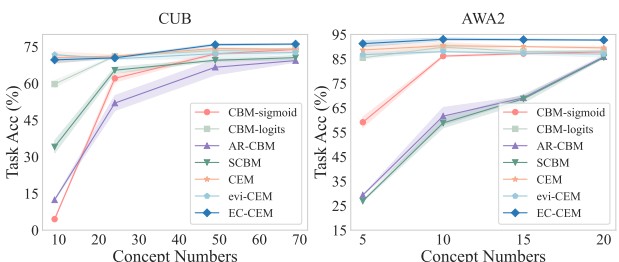

*Figure 4.* Performance comparison across varying concept numbers on the CUB and AwA2 datasets.

We further evaluate model performance using concept subsampling. Specifically, for the CUB dataset, we randomly sampled 10%, 20%, 40%, and 60% of the 28 concept groups following (Koh et al., 2020). For AwA2, we created smaller subsets from a curated group of 20 highly correlated concepts (see supplementary materials). As illustrated in Figure 4, EC-CEM demonstrates significantly higher and more stable performance than all competing methods as concepts become sparse. Specifically, the performance lead of EC-CEM over AR-CBM is particularly pronounced: on CUB (concepts reduced from 100% to 10%), classification accuracy advantage increased from **3.08% to 57.24%**; on AwA2 (concepts reduced from 85 to 10), the advantage expanded from **2.6% to 31.96%**. Notably, the widening

performance gap under sparsity provides compelling evidence for our central claim: that Copula-based dependency modeling significantly outperforms alternative approaches in handling incomplete conceptual information.

## 4.4. Quantifying Correlated Concept Correction

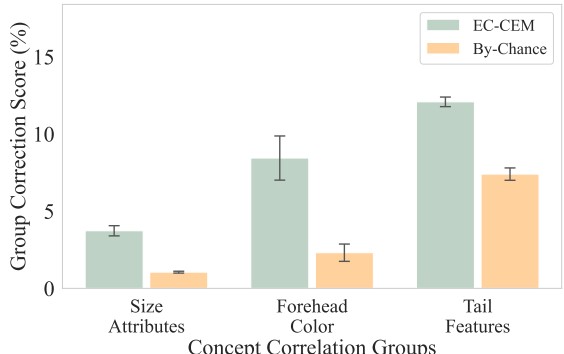

*Figure 6.* GCS under different concept groups.

Our Gaussian Copula Procedure refines initial, independent concept probabilities by modeling their underlying concept dependencies. Since such dependencies are most pronounced within semantic groups, we conduct a group-wise evaluation on the CUB dataset using the Group Correction Score (GCS), defined as the proportion of samples for which at least two concepts within the same group are simultaneously corrected (i.e., their classifications flip from

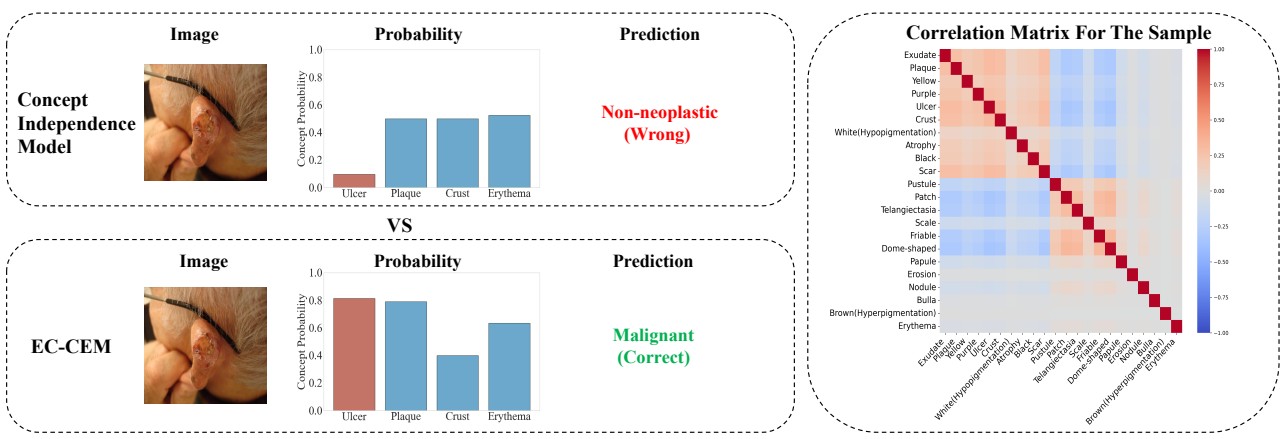

*Figure 5.* Medical diagnostic case study comparing concept independence model and our proposed method (left), with sample-derived correlation matrix showing learned concept dependencies (right).

incorrect to correct via $R$). Because co-correction is only meaningful when multiple predictions are available for recovery, we restrict the evaluation to hard samples on which the base model initially commits at least two errors within the target group. Samples with fewer initial errors mathematically preclude co-correction and would otherwise deflate the metric. To verify that the observed corrections stem from the learned dependency structure $R$ rather than from each concept's marginal accuracy, we compare against a By-Chance baseline obtained via a permutation test: we independently permute each concept's per-sample correction outcomes across samples and recompute GCS, averaging over multiple permutations. As shown in Figure 6, EC-CEM consistently exceeds this baseline across three concept groups, indicating that co-corrections are driven by the learned dependency structure $R$ rather than by chance, and that $R$ captures a generalizable dependency pattern across different types of intra-group correlations.

### 4.5. Intervention

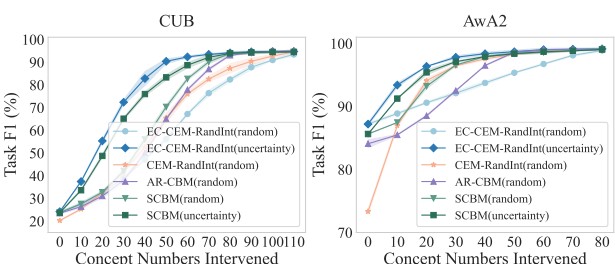

*Figure 7.* Concept intervention on the CUB and AwA2 datasets.

We adopt the CEM-RandInt training strategy on the CUB and AwA2 datasets. For the intervention policy, we compare against SCBM's uncertainty heuristic, which prioritizes concepts based on the distance to the decision boundary (i.e., $|p_i - 0.5|$). In contrast, EC-CEM leverages its evidential

Beta marginals to quantify explicit epistemic uncertainty ($u = \frac{2}{\alpha+\beta}$). This theoretically grounded metric captures evidence-based epistemic uncertainty, prioritizing concepts with insufficient evidence where the model is most uncertain. As demonstrated in Figure 7, our approach consistently outperforms baselines across both datasets, validating the superiority of evidence-based intervention.

### 4.6. Concept Alignment and Calibration

As shown in Table 2, EC-CEM consistently achieves the highest CAS and lowest ECE across all datasets. This dual improvement suggests that EC-CEM generates concept embeddings that are semantically aligned and underpinned by reliably calibrated probabilities.

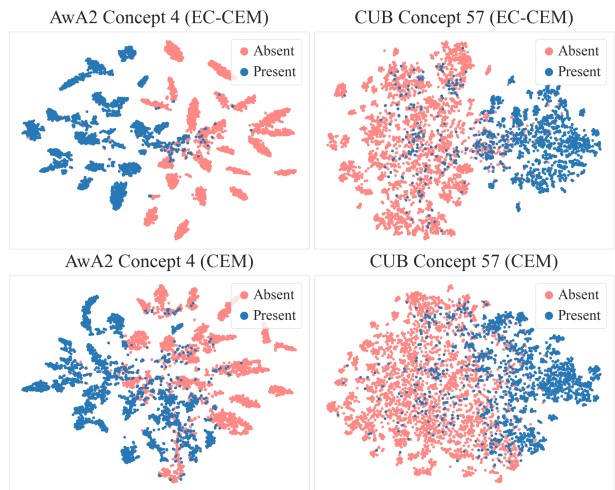

*Figure 8.* Concept embeddings t-SNE visualization.

To further validate this finding qualitatively, we employ t-SNE (Maaten & Hinton, 2008) to visualize the learned concept embedding representations $\mathbf{e}_k$. Figure 8 demonstrates

that our method achieves superior concept embedding representation compared to CEM, with markedly clearer decision boundaries. The enhanced clustering structure indicates that our approach learns more semantic concept representations in the embedding space.

*Table 2.* Comparison of CAS (↑) and Concept ECE (↓). All results are reported as mean values.

|  | CUB | | AwA2 | | SkinCon | |
| --- | --- | --- | --- | --- | --- | --- |
| Method | CAS | ECE | CAS | ECE | CAS | ECE |
| CEM | 93.46 | 6.48 | 86.34 | 3.65 | 62.95 | 5.14 |
| evi-CEM | 94.16 | 5.63 | 86.53 | 2.32 | 62.61 | 4.84 |
| EC-CEM | **94.49** | **4.39** | **91.38** | **1.99** | **63.17** | **4.35** |

### 4.7. Medical application

Figure 5 presents a medical case study comparing two concept models. While both methods identify fundamental concepts (plaque, crust, erythema), the independence model fails to capture clinically significant correlations. The correlation matrix reveals task-aware dependencies, particularly a strong crust-ulcer correlation that reflects authentic medical knowledge, as crusting frequently co-occurs with ulcerative lesions in practice. Our Copula-based approach effectively captures these clinically relevant dependencies without overfitting, achieving significantly higher ulcer detection probability than the independence baseline. These results demonstrate our method's ability to learn meaningful clinical relationships that independence-based models miss.

## 5. Conclusion and Future Work

We presented EC-CEM, a probabilistic framework that relaxes the common independence assumption in concept embedding models and decouples concept-level uncertainty from inter-concept dependency. Building on Copula theory and evidential deep learning, EC-CEM models per-concept epistemic uncertainty through evidential marginals and captures concept dependencies through a Gaussian Copula, improving interpretability and yielding strong performance, particularly in sparse-concept scenarios. A variational-inference-based optimization procedure enables tractable learning, and experiments on multiple datasets validate the effectiveness of the framework.

**Limitations and future work.** While Gaussian Copulas offer tractable inference and effectively capture concept dependencies, they have several limitations that suggest directions for future work. First, they are not well suited to heavy-tailed dependencies, which could be addressed by extending EC-CEM to richer Copula families such as Student-t Copulas. Second, the Gaussian Copula imposes a symmetric correlation structure, whereas real-world concept logic is often one-directional. For example, a hooked beak strongly implies a bird of prey, but the reverse implication is much weaker. This mismatch between a symmetric model and asymmetric semantics can introduce errors under high input uncertainty (e.g., blurry images), and capturing such asymmetry through directed dependency models is a promising direction. Finally, the procedure introduces non-trivial computational overhead, and improving its efficiency would facilitate scaling to larger concept sets.

## Acknowledgments

This work was supported by the National Natural Science Foundation of China ( Nos. 62476165, 62472315, 62406182) and the Science and Technology Commission of Shanghai Municipality (No. 25511102102).

## Impact Statement

This paper presents work whose goal is to advance the field of Machine Learning. There are many potential societal consequences of our work, none which we feel must be specifically highlighted here.

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

## A. ELBO Derivation

We formulate our training objective through variational inference and provide a complete theoretical proof in this section.

### A.1. Expected concept likelihood

Let $\mathbb{E}_q[\cdot] \equiv \mathbb{E}_{q(\boldsymbol{\mu}|\mathbf{h})}[\cdot]$ and $\ell_k(\mu_k) = c_k \log \mu_k + (1 - c_k) \log(1 - \mu_k)$.

$$
\begin{aligned}
\mathbb{E}_q[\log p(\mathbf{c}|\boldsymbol{\mu})] = \mathbb{E}_q \left[ \sum_{k=1}^{K} \ell_k(\mu_k) \right] \\
= \sum_{k=1}^{K} \mathbb{E}_q[\ell_k(\mu_k)] \\
= \sum_{k=1}^{K} \mathbb{E}_{Beta_{\theta_k^q}(\mu_k|\mathbf{x})}[\ell_k(\mu_k)] \\
= \sum_{k=1}^{K} [(1 - c_k)\psi(\beta_k) - \psi(\alpha_k + \beta_k) + c_k \psi(\alpha_k)].
\end{aligned}
\tag{18}
$$

### A.2. KL-divergence term

For brevity, let $q \equiv q(\boldsymbol{\mu}|\mathbf{h})$ (posterior), $p \equiv p(\boldsymbol{\mu})$ (prior), $c_{\mathbf{R}}$ denote the copula density, and $f_k^q$, $f_k^p$ the marginal densities with posterior and prior parameters.

$$
\begin{aligned}
\mathrm{KL}[q\|p] = \int q \log \frac{q}{p} d\boldsymbol{\mu} \\
= \int q \left[ \log c_{\mathbf{R}} + \sum_k \log f_k^q - \sum_k \log f_k^p \right] d\boldsymbol{\mu} \\
= \int q \log c_{\mathbf{R}} d\boldsymbol{\mu} + \sum_k \int q_k [\log f_k^q - \log f_k^p] d\mu_k \\
= \underbrace{\mathbb{E}_{c_{\mathbf{R}}}[\log c_{\mathbf{R}}]}_{\text{Copula Reg.}} + \underbrace{\sum_{k=1}^{K} \mathrm{KL}[Beta(\alpha_k^q, \beta_k^q)\|Beta(\alpha_k^p, \beta_k^p)]}_{\text{Marginal Reg.}}
\end{aligned}
\tag{19}
$$

**Copula Dependency Regularization** The p.d.f. of Gaussian Copula is:

$$
c_{\mathbf{R}}(\mathbf{u}) = \frac{1}{\sqrt{\det \mathbf{R}}} \exp\left( -\frac{1}{2} \mathbf{z}^T (\mathbf{R}^{-1} - \mathbf{I})\mathbf{z} \right)
$$

where $\mathbf{z} = [\Phi^{-1}(u_1), \ldots, \Phi^{-1}(u_K)]^T \sim \mathcal{N}(0, \mathbf{R})$ and $\mathbf{u} = [\Phi(z_1), \ldots, \Phi(z_K)]^T$. The Copula term can be derived as:

$$
\mathbb{E}_{c_{\mathbf{R}}} \log(c_{\mathbf{R}}(\mathbf{u})) = -\frac{1}{2} \log \det(\mathbf{R}) - \frac{1}{2} \mathbb{E}[\mathbf{z}^\top (\mathbf{R}^{-1} - \mathbf{I})\mathbf{z}].
$$

For the right part of the formula:

$$\begin{aligned}
\mathbb{E}[\mathbf{z}^\top(\mathbf{R}^{-1} - \mathbf{I})\mathbf{z}] &= \mathbb{E}[\text{tr}(\mathbf{z}^T(\mathbf{R}^{-1} - \mathbf{I})\mathbf{z})] \\
&= \mathbb{E}[\text{tr}((\mathbf{R}^{-1} - \mathbf{I})\mathbf{z}\mathbf{z}^T)] \\
&= \text{tr}((\mathbf{R}^{-1} - \mathbf{I})\mathbb{E}(\mathbf{z}\mathbf{z}^T)) \\
&= \text{tr}(\mathbf{R}^{-1}\mathbf{R}) - \text{tr}(\mathbf{I}\mathbf{R}) \\
&= \text{tr}(\mathbf{I}) - \text{tr}(\mathbf{R}) = 0,
\end{aligned}$$

where $\mathbf{R}$ is the concept correlation matrix.

We can find that the $\text{KL}[\mathcal{N}(\mathbf{0}, \mathbf{R})\|\mathcal{N}(\mathbf{0}, \mathbf{I})]$ is equivalent to $\mathbb{E}_{c_\mathbf{R}} \log(c_\mathbf{R}(\mathbf{u}))$.

$$\begin{aligned}
\text{KL}[\mathcal{N}(\mathbf{0}, \mathbf{R})\|\mathcal{N}(\mathbf{0}, \mathbf{I})] &= \frac{1}{2}\left(\text{tr}(\mathbf{R}) - K - \log \det \mathbf{R}\right) \\
&= -\frac{1}{2}\log \det(\mathbf{R}).
\end{aligned}$$

$$\begin{aligned}
\mathbb{E}_{c_\mathbf{R}} \log(c_\mathbf{R}(\mathbf{u})) &= -\frac{1}{2}\log \det(\mathbf{R}) - \frac{1}{2}\mathbb{E}[\mathbf{z}^\top(\mathbf{R}^{-1} - \mathbf{I})\mathbf{z}] \\
&= -\frac{1}{2}\log \det(\mathbf{R}) \\
&= \text{KL}[\mathcal{N}(\mathbf{0}, \mathbf{R})\|\mathcal{N}(\mathbf{0}, \mathbf{I})].
\end{aligned}$$

**Marginal KL Regularization** We interpret the marginal KL term as a penalty (Han et al., 2022; Gao et al., 2024). Then the marginal KL defined as:

$$\text{KL}_{\text{Marginal}} \approx \sum_{k=1}^{K} \text{KL}[\text{Beta}(\tilde{\alpha}_k^q, \tilde{\beta}_k^q)\|\text{Beta}(1, 1)],$$

where the parameterization $\tilde{\alpha}_k^q = c_k + (1 - c_k)\alpha_k^q$ and $\tilde{\beta}_k^q = c_k\beta_k^q + (1 - c_k)$. This marginal KL term penalizes evidence for incorrect predictions but preserves learned parameters for correct ones, while the Copula term penalizes redundant dependencies, thereby reducing over-fitting by preventing the model from capturing needlessly intricate correlations. We demonstrate the Marginal KL calculation through two boundary cases that provide clear mathematical illustrations of the underlying derivation process as follows:

When $c_k = 1$, then $\hat{\alpha}_k^q = 1$ and $\hat{\beta}_k^q = \beta_k^q$, the KL term can be derived as:

$$\text{KL}[\text{Beta}(\hat{\alpha}_k^q, \hat{\beta}_k^q)\|\text{Beta}(1, 1)] = \log(\beta_k^q) + \frac{1 - \beta_k^q}{\beta_k^q}.$$

When $c_k = 0$, then $\hat{\alpha}_k^q = \alpha_k^q$ and $\hat{\beta}_k^q = 1$, the KL term can be derived as:

$$\text{KL}[\text{Beta}(\hat{\alpha}_k^q, \hat{\beta}_k^q)\|\text{Beta}(1, 1)] = \log(\alpha_k^q) + \frac{1 - \alpha_k^q}{\alpha_k^q}.$$

We approximate the marginal KL with the evidence-based penalty used in EDL:

$$\begin{aligned}
\text{KL}\big[q(\boldsymbol{\mu}|\mathbf{h})\|p(\boldsymbol{\mu})\big] &= \underbrace{\text{KL}[\mathcal{N}(\mathbf{0}, \mathbf{R})\|\mathcal{N}(\mathbf{0}, \mathbf{I})]}_{\text{Copula Dependency Reg.}} + \underbrace{\sum_{k=1}^{K} \text{KL}[\text{Beta}(\hat{\alpha}_k^q, \hat{\beta}_k^q)\|\text{Beta}(1, 1)]}_{\text{Marginal Reg.}} \\
&= -\frac{1}{2}\log \det(\mathbf{R}) + \sum_{k=1}^{K}\left(c_k[\log(\beta_k^q) + \frac{1 - \beta_k^q}{\beta_k^q}] + (1 - c_k)[\log(\alpha_k^q) + \frac{1 - \alpha_k^q}{\alpha_k^q}]\right). \quad (20)
\end{aligned}$$

**Total loss function** The final loss function is formulated as:

$$\mathcal{L} = \mathbb{E}_{q(\boldsymbol{\mu}|\mathbf{h})}\left[-\log p(\mathbf{c}|\boldsymbol{\mu})\right] + \lambda_1 \cdot \mathbb{E}_{q(\boldsymbol{\mu}|\mathbf{h})}\left[-\log p(\mathbf{y}|\boldsymbol{\mu}, \mathbf{h})\right] + \lambda_2 \cdot \mathrm{KL}\left[q(\boldsymbol{\mu}|\mathbf{h}) \parallel p(\boldsymbol{\mu})\right]. \tag{21}$$

where $\lambda_1$ represents the balance coefficient that weights the trade-off between the downstream task objective and the concept prediction objective and $\lambda_2$ is the annealing coefficient that controls the balance between the likelihood and prior to prevent posterior collapse. We decompose the regularization parameter $\lambda_2$ into two distinct hyperparameters: $\lambda_{marginal}$ for marginal regularization and $\lambda_{copula}$ for Copula dependency regularization, enabling independent control over these complementary regularization mechanisms.

## B. Comparisons with Prior Approaches

In this section, we compare our proposed EC-CEM with representative concept bottleneck methods that explicitly model concept dependencies, as summarized in Table 3. We focus on two key methodological distinctions: (i) the structural decoupling of uncertainty and correlation, and (ii) the topology of the modeling space.

### B.1. Distribution-Level Decoupling of Uncertainty and Correlation

A common limitation of logit-space dependency modeling (e.g., SCBM (Vandenhirtz et al., 2024)) is that both *per-concept predictive dispersion* and *inter-concept dependency* are parameterized within the same latent Gaussian space. Specifically, SCBM models concept logits $\mathbf{z}$ via a multivariate normal distribution $\mathbf{z} \sim \mathcal{N}(\boldsymbol{\mu}, \boldsymbol{\Sigma})$, whose covariance matrix $\boldsymbol{\Sigma}$ jointly determines the variability of individual concept logits and their statistical dependencies.

This shared parameterization creates a mathematical *entanglement* between uncertainty and correlation. Consider the covariance matrix $\boldsymbol{\Sigma}$ and the resulting correlation coefficient $\rho_{ij}$ between two concepts $i$ and $j$:

$$\boldsymbol{\Sigma} = \begin{bmatrix} \sigma_i^2 & \sigma_{ij} \\ \sigma_{ij} & \sigma_j^2 \end{bmatrix}, \quad \rho_{ij} = \frac{\sigma_{ij}}{\sqrt{\sigma_i^2 \cdot \sigma_j^2}}. \tag{22}$$

In this framework, the diagonal term $\sigma_i^2$ (variance) captures the predictive dispersion of concept $i$ in the logit space; a larger variance typically yields a broader (i.e., less confident) predictive distribution after the sigmoid mapping. However, this formulation introduces an inherent coupling: when the model increases $\sigma_i^2$ to reflect ambiguous evidence for concept $i$ (e.g., due to occlusion or blur), the denominator in $\rho_{ij}$ also increases. Consequently, *unless the covariance term $\sigma_{ij}$ scales proportionally*, the implied correlation coefficient $\rho_{ij}$ will decrease. While the network can, in principle, learn to adjust $\sigma_{ij}$ accordingly, this imposes an implicit constraint: maintaining stable correlations under varying uncertainty requires coordinated adaptation of both variance and covariance parameters, rather than independent control.

This may lead to an undesirable distortion in the model's expressed dependency: uncertainty about a concept can inadvertently weaken the *statistical* correlation the model attributes to that concept with others. Importantly, this does not mean the underlying semantic relationship disappears—for example, the fact that black legs imply a specific plumage pattern remains true regardless of observation quality. Rather, under Gaussian logit modeling, these two attributes—marginal predictive dispersion and dependency strength—are tightly coupled through $\boldsymbol{\Sigma}$.

In contrast, our Copula-EDL framework decouples uncertainty and dependency **at the distributional level by construction**:

- **Uncertainty (Marginals):** Each concept is modeled with a Beta predictive distribution parameterized by evidence $(\alpha, \beta)$, providing an explicit and interpretable measure of predictive uncertainty naturally supported on $[0, 1]$.

- **Correlation (Dependency):** Concept dependencies are modeled separately via a Gaussian copula defined in the cumulative distribution space, with its correlation parameters learned independently of the marginal evidence.

This separation provides a *structural guarantee* that marginal uncertainty does not directly rescale the dependency parameters, making uncertainty and correlation independently interpretable and controllable—without relying on the network to implicitly learn such invariance.

## B.2. Dependency Modeling in Probability Space

Table 3 further highlights a fundamental difference in the **modeling space**. Logit-based approaches operate in an unbounded domain $(-\infty, +\infty)$ and rely on a Sigmoid mapping to obtain probabilities in $[0, 1]$. Due to the saturation behavior of the Sigmoid function, variations in logits may be strongly attenuated when probabilities are near 0 or 1, which can make probability-level variability less sensitive under highly confident predictions. As a result, dependencies expressed in logit space may become harder to preserve and interpret in probability space.

In comparison, EC-CEM models dependencies directly on bounded predictive distributions within $[0, 1]$. This topology aligns with the probabilistic nature of concepts and avoids additional post-transformations that may obscure probability-level dependency patterns.

## B.3. Flexibility via Marginal–Dependency Separation

A core theoretical advantage of Copula-based modeling is the separation between *marginal distributions* and the *dependency structure*. Unlike latent Gaussian formulations that implicitly impose modeling assumptions in logit space, the Copula framework allows choosing marginals that best characterize the data while retaining a dedicated dependency model.

We leverage this flexibility by adopting Beta marginals for concept prediction. Compared to Gaussian approximations, Beta distributions are well-suited for bounded probabilities: they can represent confident unimodal distributions, uncertain near-uniform distributions (low evidence), and even U-shaped (bimodal) behaviors. This enables EC-CEM to capture diverse predictive characteristics that are difficult to represent robustly with rigid logit-space parameterizations. Furthermore, while we employ a Gaussian copula for computational efficiency, the framework is generic and can be extended to richer copula families when more complex dependency patterns (e.g., tail dependence) are needed.

*Table 3.* Comparison between AR-CBM, SCBM and EC-CEM

| Aspect | AR-CBM | SCBM | EC-CEM (Ours) |
|---|---|---|---|
| Modeling Space | Probability space | Logit space | Evidential CDFs (copula space) |
| Dependency Modeling | Autoregression | Multivariate Normal (logits) | Gaussian Copula |
| Theoretical Foundation | Classifier chains | Logistic-normal modeling | EDL + Copula theory |
| Uncertainty Quantification | - 
 - | Logit dispersion (via $\boldsymbol{\Sigma}$) 
 Intervention heuristic: $\lvert p(c_k) - 0.5 \rvert$ | Explicit epistemic uncertainty 
 $u_k = \frac{2}{\alpha_k + \beta_k}$ |

# C. Scalability to Large Concept Sets

**Scaling to very large $K$.** In many concept-based datasets, the number of human-defined concepts is on the order of tens to a few hundreds, for which an $O(K^2)$ copula parameterization is practically affordable. For applications with substantially larger concept vocabularies (e.g., thousands of concepts), EC-CEM can be extended with structured copula parameterizations such as:

- block-diagonal or group-wise copulas, where concepts are partitioned into semantically coherent groups and a separate copula is learned per group;

- sparse or low-rank factorizations of the copula covariance, reducing both storage and computation;

- hierarchical or vine copulas that decompose high-dimensional dependencies into collections of lower-dimensional copulas.

These extensions are orthogonal to the core evidential copula formulation proposed in this work and provide a natural path for scaling EC-CEM to very large concept sets while preserving the core evidential copula framework.

# D. Additional Experiments

## D.1. AwA2 Concept Correlation Matrix in Abaltion Study

Figure 9 shows concept correlation matrix of the 20 curated concepts from the AwA2 dataset. Correlation is measured by the Pearson correlation coefficient, calculated across the 50 animal classes. To reveal the dependency structure, the concepts are reordered using hierarchical clustering. The clustering was performed using Ward's linkage method on a distance matrix defined as $1 - |\text{correlation}|$. The resulting block-diagonal structure visually confirms the presence of several clusters of highly inter-correlated concepts, such as aquatic and carnivore-related attributes.

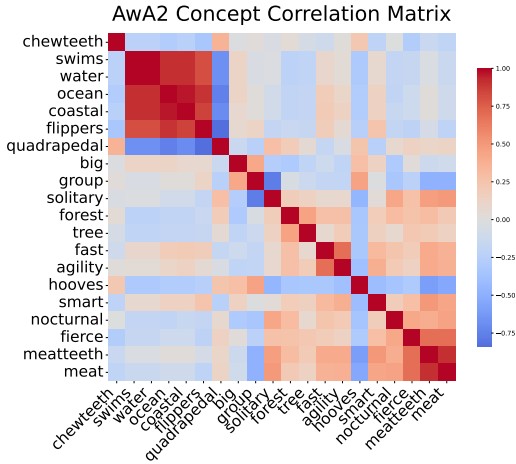

*Figure 9.* AwA2 Concept Ablation study Concept Correlation matrix.

## D.2. Semantic Concept Embedding Visualizations

To provide qualitative analysis of the learned concept representations, we employ t-SNE visualization to compare the concept embeddings produced by our proposed EC-CEM method against the baseline CEM approach. We evaluate this visualization on two benchmark datasets: AwA2 and CUB.

Figure 10 and Figure 11 demonstrate that EC-CEM achieves significantly improved class separability compared to CEM. The visualizations reveal that our method produces more distinct and well-defined decision boundaries between different classes, indicating superior concept disentanglement and representation quality. This enhanced separability suggests that EC-CEM learns more discriminative concept embeddings that better capture the underlying semantic structure of the data.

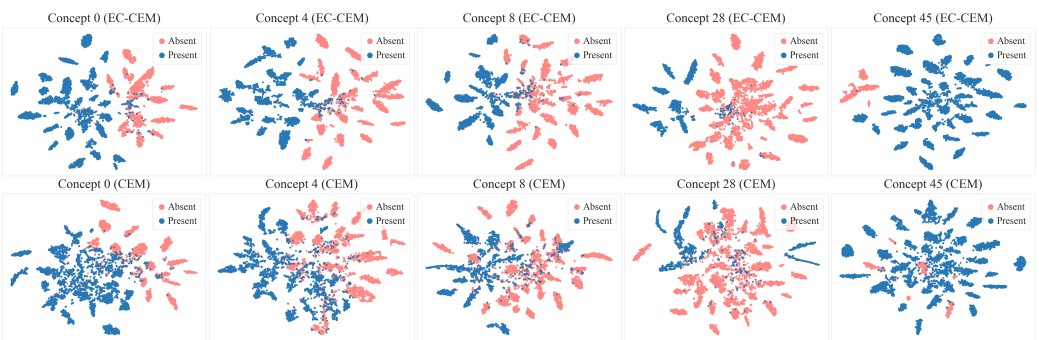

*Figure 10.* Concept Embedding visualization on the AwA2 dataset using t-SNE

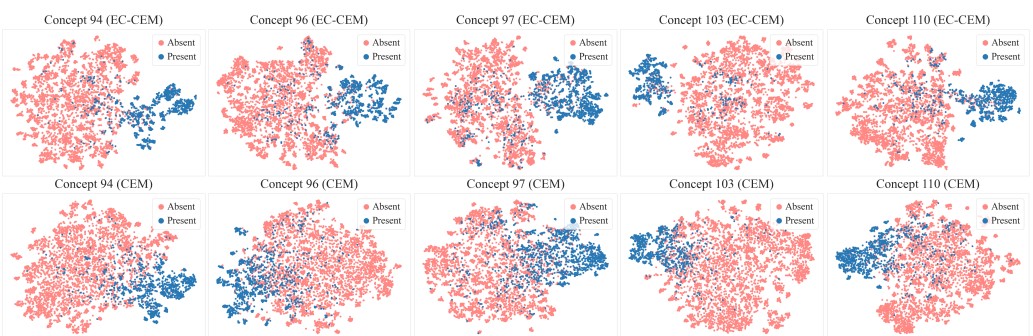

*Figure 11.* Concept Embedding visualization on the CUB dataset using t-SNE

# E. Implementation Details

## E.1. Detailed dataset settings

**CUB:** The data augmentation pipeline, following CEM, incorporates color jittering to randomly adjust brightness and saturation, random resized cropping and horizontal flipping. The hyperparameter $\lambda_1$ is set to 1 and the learning rate is set to $10^{-4}$. $\lambda_{marginal}$ follows an adaptive schedule $\lambda_{marginal} = \min(1, \tau/20)$ where $\tau$ denotes the current epoch number and $\lambda_{copula}$ follows a step schedule that activates at epoch 50, setting $\lambda_{copula} = 1$ for $\tau \geq 50$ and $\lambda_{copula} = 0$ otherwise. We set the number of Monte Carlo samples to be 15, as we found 15 samples is sufficient to get a stable estimator.

**SkinCon** The data augmentation pipeline, following evi-CEM, applies random resized cropping, rotation, color jittering, and horizontal flipping before center cropping. The hyperparameter $\lambda_1$ is set to 1. The learning rate is set to $5 \times 10^{-4}$. $\lambda_{marginal}$ follows an adaptive schedule $\lambda_{marginal} = \min(1, \tau/30)$ and $\lambda_{copula}$ follows a step schedule that activates at epoch 40, setting $\lambda_{copula} = 1$ for $\tau \geq 40$ and $\lambda_{copula} = 0$ otherwise. We set the number of Monte Carlo samples to be 15, as we found 15 samples is sufficient to get a stable estimator.

**AwA2** The data augmentation pipeline implements standard geometric transformations by resizing images to 256×256 pixels, applying random cropping to 224×224 resolution, and incorporating random horizontal flipping for spatial variation. The hyperparameter $\lambda_1$ follows an adaptive schedule $\lambda_1 = \min(1, \tau/10)$. The learning rate is set to $5 \times 10^{-5}$. $\lambda_{marginal}$ follows an adaptive schedule $\lambda_{marginal} = \min(1, \tau/10)$ and $\lambda_{copula}$ follows a step schedule that activates at epoch 25, setting $\lambda_{copula} = 1$ for $\tau \geq 25$ and $\lambda_{copula} = 0$ otherwise. We set the number of Monte Carlo samples to be 15, as we found 15 samples is sufficient to get a stable estimator.

## E.2. Beta pytorch implementation

Our framework implements the Beta cumulative distribution function (Beta$_{CDF}$) and inverse cumulative distribution function (Beta$_{ICDF}$) using PyTorch. For Beta$_{CDF}$, we employ Gauss-Legendre numerical integration with 128 nodes to accurately approximate the integral. For Beta$_{ICDF}$, we utilize the Newton-Raphson iterative method, setting a convergence tolerance of $1 \times 10^{-6}$ and a maximum of 20 iterations to ensure numerical stability and convergence. To validate the accuracy of our custom Beta distribution implementations, we compared them against the `scipy.stats` beta distribution functions for parameter values where $\alpha, \beta \geq 1$. Specifically, we tested across a comprehensive range of parameters with $\alpha \in \{1, 2, 5, 10, 20, 50\}$ and $\beta \in \{1, 3, 8, 15, 25, 40\}$. The results demonstrate that our Beta$_{CDF}$ implementation achieves excellent accuracy with a mean relative error of $2.43 \times 10^{-6}$, and a median relative error of $6.73 \times 10^{-7}$. For the Beta$_{ICDF}$, while maintaining good overall accuracy, we observed a mean relative error of $1.59 \times 10^{-4}$, and a median relative error of $2.62 \times 10^{-7}$. The notably low median errors for both implementations indicate robust performance across most parameter combinations.

## E.3. Concept Alignment Score

**Concept Alignment Score (CAS)** evaluates the trustworthiness of learned concept representations by measuring the homogeneity of predicted concept labels within similar samples in concept space. For each concept $c_i$, we apply clustering to group samples using their concept representations $\{\hat{c}_i^{(1)}, \hat{c}_i^{(2)}, \dots\}$. The homogeneity score $h(\cdot)$ computes the conditional entropy between ground truth labels and cluster assignments. Higher homogeneity indicates $h(\cdot)$ better alignment between

learned representations and true concept labels. CAS averages these scores across all concepts and cluster numbers, yielding a normalized score in $[0, 1]$:

$$\text{CAS}(\hat{c}_1, \ldots, \hat{c}_k) = \frac{1}{N-2} \sum_{\rho=2}^{N} \left( \frac{1}{k} \sum_{i=1}^{k} h(C_i, \Pi_i(\kappa, \rho)) \right)$$

In practice, we use k-Medoids clustering and vary $\rho$ with step size $\delta > 1$ for computational efficiency. This metric performs clustering-based homogeneity analysis by comparing learned representations against ground truth labels, providing a robust measure of how well the model captures meaningful semantic relationships in the concept space.

### E.4. Grouping Strategy for CUB dataset

We employ a multi-level hierarchical grouping approach that organizes concepts according to three primary organizing principles: anatomical locality, attribute type consistency, and functional relatedness. Each concept may belong to multiple groups across different hierarchical levels, enabling analysis of improvement patterns at various levels of semantic granularity.

#### E.4.1. ANATOMICAL GROUPING

The anatomical grouping strategy partitions concepts based on their association with distinct body regions of avian anatomy. This approach reflects the biological organization of bird morphology and captures spatial relationships between visual features.

**Anatomical Groups Definition:**

- **Bill Features**: $G_{\text{bill}} = \{c \in C : \text{'bill'} \subset c\}$

- **Wing Features**: $G_{\text{wing}} = \{c \in C : \text{'wing'} \subset c\}$

- **Tail Features**: $G_{\text{tail}} = \{c \in C : \text{'tail'} \subset c\}$

- **Head Features**: $G_{\text{head}} = \{c \in C : \exists x \in \{\text{'head', 'crown', 'forehead', 'nape'}\}, x \subset c\}$

- **Chest-Belly Features**: $G_{\text{chest}} = \{c \in C : \exists x \in \{\text{'breast', 'belly', 'underparts'}\}, x \subset c\}$

- **Back Features**: $G_{\text{back}} = \{c \in C : \exists x \in \{\text{'back', 'upperparts'}\}, x \subset c\}$

- **Throat Features**: $G_{\text{throat}} = \{c \in C : \text{'throat'} \subset c\}$

- **Leg Features**: $G_{\text{leg}} = \{c \in C : \text{'leg'} \subset c\}$

- **Eye Features**: $G_{\text{eye}} = \{c \in C : \text{'eye'} \subset c\}$

where $C$ represents the complete set of concept names and $\subset$ denotes substring containment.

#### E.4.2. CONCEPT TYPE GROUPING

The Concept type grouping organizes concepts according to the nature of the visual property they describe, independent of anatomical location. This categorization enables analysis of how different types of visual Concepts respond to the enhancement process.

**Concept Type Groups Definition:**

- **Color Features**: $G_{\text{color}} = \{c \in C : \text{'color'} \subset c\}$

- **Pattern Features**: $G_{\text{pattern}} = \{c \in C : \text{'pattern'} \subset c\}$

- **Shape Features**: $G_{\text{shape}} = \{c \in C : \text{'shape'} \subset c\}$

- **Size Features**: $G_{\text{size}} = \{c \in C : \text{'size'} \subset c\}$

- **Length Features**: $G_{\text{length}} = \{c \in C : \text{'length'} \subset c\}$

### E.4.3. FUNCTIONAL GROUPING

The functional grouping strategy combines anatomically related regions that serve similar biological functions or exhibit strong visual coherence. This approach captures higher-level semantic relationships that may not be apparent from purely anatomical considerations.

**Functional Groups Definition:**

- **Cephalic Region**: $G_{\text{Cephalic}} = \{c \in C : \exists x \in \{\text{'head', 'crown', 'forehead', 'nape', 'throat'}\}, x \subset c\}$

- **Axial Body**: $G_{\text{Axial}} = \{c \in C : \exists x \in \{\text{'breast', 'belly', 'back', 'upperparts', 'underparts'}\}, x \subset c\}$

- **Locomotory Appendages**: $G_{\text{Locomotory}} = \{c \in C : \exists x \in \{\text{'wing', 'tail'}\}, x \subset c\}$

- **Feeding Apparatus**: $G_{\text{feeding}} = \{c \in C : \exists x \in \{\text{'bill', 'eye'}\}, x \subset c\}$

- **Support Apparatus**: $G_{\text{support}} = \{c \in C : \text{'leg'} \subset c\}$

