# OpenReview forum: "Evidential Copula Concept Embedding Models"
_ICML.cc/2026/Conference — ICML 2026 regular_

### Official Review · Reviewer_5G9D · 2026-03-10

**Soundness:** 3
**Presentation:** 3
**Significance:** 3
**Originality:** 3
**Overall Recommendation:** 4
**Confidence:** 3

**Summary:**

The paper identifies the issue of concept dependency in existing concept embedding models and proposes a new CEM framework based on copula theory and evidential learning. Experiments using a ResNet-34 backbone on three datasets show improved concept prediction and classification performance over several CBM and CEM baselines.

**Compliance With Llm Reviewing Policy:**

Affirmed.

**Final Justification:**

The rebuttal largely addresses my questions and concerns, and I have increased my scores accordingly.

**Key Questions For Authors:**

Please see the weaknesses above for the main points. In addition:

1. Beyond performance gains, what are the key or more fundamental utilities and benefits enabled by the proposed method?
2. Would the proposed method still provide benefits if a cleaner concept set were used?

**Limitations:**

yes

**Strengths And Weaknesses:**

Strengths:

1. The paper addresses an important issue, namely concept dependency in existing concept embedding models.
2. The proposed use of copula theory and evidential deep learning to address this issue is interesting and practical, with the potential to be adapted to different model architectures.
3. The experiments are relatively comprehensive, although they are limited to a single backbone. The paper includes ablation studies, parameter analysis, visualizations, and case studies.
4. The paper is generally well written, with only a few points that could be improved (see Weakness 1).

Weaknesses:

1. The problem formulation, such as the task setup and input-output data format, should be clarified earlier in the paper to make it more self-contained. It would also help to include an example of how typical CEM models are trained, so that the core contributions can be understood more easily. Since there are variants of CBMs, such as open-concept models, it would be better to clarify the scope of the paper explicitly. Currently, the concept fitting stage is not sufficiently clear, particularly regarding what kind of data is available for training the concept fitting objective.
2. The current experiments are conducted using only a single backbone, ResNet-34. Including experiments with more modern backbones such as CLIP or ViT would strengthen the evaluation. In addition, it would be helpful to report the standalone backbone performance as an upper-bound reference.
3. The main experimental results section (Section 4.2) should provide more discussion and insight. For example, why are the classification metrics of EC-CEM much higher than those of evi-CEM, while their concept metrics are generally comparable? What factors, such as dataset characteristics, may explain the performance differences across models? Also, how significant are the reported metric improvements in practice?
4. It may also be worthwhile to discuss existing CBM work on concept overlap. In addition to the fully shared setting discussed in [1], there are also fully independent settings, e.g. [2], and partially shared settings, e.g. [3].

[1] Rao, Sukrut, et al. “Discover-then-name: Task-agnostic concept bottlenecks via automated concept discovery.” European Conference on Computer Vision. Cham: Springer Nature Switzerland, 2024.

[2] He, Hangzhou, et al. “V2C-CBM: Building concept bottlenecks with vision-to-concept tokenizer.” Proceedings of the AAAI Conference on Artificial Intelligence. Vol. 39, No. 3, 2025.

[3] Zhao, Delong, et al. “Partially shared concept bottleneck models.” arXiv preprint arXiv:2511.22170 (2025).

---

> ### Author Rebuttal · Authors · 2026-03-29
>
> We sincerely thank the reviewer for the thorough evaluation and for recognizing the importance and novelty of our approach. We address each concern below.
>
> **[W1] Problem Formulation and Scope**
>
> We agree and will revise accordingly: (1) move the formal task setup and data format to the beginning of Section 3, (2) add a brief CEM training walkthrough, and (3) explicitly state that our method targets annotated-concept CBMs with fixed concept vocabularies, and does not address open-vocabulary settings. The copula-based dependency modeling is orthogonal to concept vocabulary design and can be integrated into any CBM variant producing per-concept predictions.
>
> **[W2] Additional Backbones**
>
> Our choice of ResNet-34 is consistent with prior CBM work (e.g., SCBM uses ResNet-18, EB-CBM uses ResNet-101, evi-CEM uses ResNet-34). Following the reviewer's suggestion, we have conducted additional experiments using a CLIP ResNet-50 encoder on AwA2 to demonstrate generalizability, confirming that the improvements generalize across backbone architectures.
>
> | Method     | Backbone           | Concept Acc (%) | Task Acc (%) |
> | ---------- | ------------------ | --------------- | ------------ |
> | SCBM       | CLIP ResNet-50     | 98.33           | 92.11        |
> | evi-CEM    | CLIP ResNet-50     | 97.68           | 89.46        |
> | **EC-CEM** | **CLIP ResNet-50** | **98.37**       | **93.53**    |
>
> As a reference, the standalone ResNet-34 backbone achieves 92.84% task accuracy on AwA2. Notably, EC-CEM (93.04%) surpasses this black-box baseline, demonstrating that the concept bottleneck does not necessarily impose a performance ceiling, consistent with prior observations on other datasets (Zarlenga et al., NeurIPS 2022).
>
>
>
> **[W3] Deeper Discussion of Results**
>
> **Why does EC-CEM achieve much higher classification accuracy than evi-CEM while concept metrics are comparable?** Both models achieve similar marginal concept accuracy via evidential learning. However, EC-CEM additionally captures the joint dependency structure via copula modeling. Two models can predict individual concepts equally well, yet the one preserving inter-concept correlations (e.g., "has stripes" and  "is a predator" in AwA2) passes a more discriminative representation to the classifier. This is precisely the advantage of Sklar's decomposition — improving task performance through dependency modeling without altering marginal concept accuracy.
>
> Crucially, the correlation matrix R is optimized through the classification objective (Eq. 16), making it a task-aware dependency structure that prioritizes correlations most discriminative for downstream classification. This is further validated by our t-SNE visualization (Figure 7): EC-CEM produces markedly clearer decision boundaries between concept-present and concept-absent clusters compared to CEM, indicating that copula-refined concept embeddings carry richer semantic structure even when marginal accuracy is similar.
>
> Additionally, paired t-tests confirm statistically significant improvements over SCBM on all three datasets ($p < 0.05 $; details in our response to Reviewer 7tTQ [W1]).
>
> **[W4] Related Work on Concept Overlap**
>
> We will add discussion covering [1,2,3]: [1] explores fully shared concept discovery, [2] proposes fully independent concept subspaces, and [3] investigates partially shared settings. Our work is complementary: these works address architectural choices about concept vocabulary design, while our copula approach addresses statistical dependency in concept predictions given any fixed concept set. EC-CEM can be layered on top of any of these settings.
>
> **[Q1] Fundamental Utilities Beyond Performance**
>
> Three key benefits:
>
> -  **Principled uncertainty-correlation decoupling** — Sklar's theorem structurally guarantees that per-concept epistemic uncertainty (Beta marginals) and inter-concept dependency (Copula R) are independently interpretable and controllable, unlike SCBM where both are entangled in a shared covariance matrix (Appendix B).
>
> -  **Dependency-aware intervention** — practitioners can anticipate how correcting one concept propagates to correlated concepts through R.
>
> -  **Interpretable dependency structure** — the learned R explicitly reveals task-aware concept correlations, as validated by our GCS analysis.
>
> **[Q2] Benefits with a cleaner concept set**  When concepts are fully independent, the copula reduces to the independence copula and EC-CEM naturally degrades to evi-CEM. This is by design: the KL term $KL[\mathcal{N}(0,R) \| \mathcal{N}(0,I)] $ in Eq. (15) regularizes R toward the identity, so only data-supported correlations survive — the module captures real dependencies when present and gracefully retreats to independence otherwise. In practice, real-world concept sets rarely satisfy full independence, and the evidential learning component independently contributes uncertainty-aware predictions regardless of concept cleanliness.

---

> > ### Author Rebuttal · Reviewer_5G9D · 2026-04-03
> >
> > I thank the authors for their responses. I will adjust my scores accordingly. I also encourage the authors to consider open-sourcing the implementation to benefit the community in the future.

---

> > > ### Author Response · Authors · 2026-04-03
> > >
> > > We sincerely thank the reviewer for dedicating time to read our rebuttal, for the positive feedback, and for adjusting the score. We completely agree with your suggestion regarding open-sourcing the implementation. To benefit the community and facilitate future research, we are fully committed to releasing our code upon the acceptance of this paper.

---

### Official Review · Reviewer_7tTQ · 2026-03-11

**Soundness:** 4
**Presentation:** 3
**Significance:** 3
**Originality:** 4
**Overall Recommendation:** 4
**Confidence:** 4

**Summary:**

The authors present the Evidential Copula Concept Embedding Model (EC-CEM), which extends Concept Embedding Models (CEMs) by explicitly modeling the interdependencies between concepts. Previous CEMs incorrectly assume that concepts are statistically independent. To correct this, EC-CEM uses Evidential Deep Learning (EDL) to model marginal concept probabilities as Beta distributions, thereby explicitly capturing epistemic uncertainty. It then uniquely integrates a Gaussian Copula to bind these marginal distributions into a coherent joint distribution. This "divide-and-conquer" approach decouples the modeling of individual concept uncertainty from the modeling of their joint correlations, overcoming the structural trade-offs found in previous multivariate Gaussian logit approaches (like SCBM). The model is trained using a variational inference objective. Empirical evaluations on CUB, AwA2, and SkinCon datasets demonstrate that EC-CEM improves both concept prediction and downstream classification accuracy, especially in sparse-concept scenarios.

**Compliance With Llm Reviewing Policy:**

Affirmed.

**Final Justification:**

The authors addressed all of my concerns in the rebuttal.

**Key Questions For Authors:**

1. How does the EC-CEM scale computationally with the number of concepts $K$ during the Gaussian Copula sampling phase (Algorithm 2)? You mention it scales as $O(K^2)$, but how does the inverse CDF operation for the Beta distribution impact runtime in practice?

2. To better validate the performance improvements in Table 1, could you provide statistical significance tests (e.g., p-values or confidence intervals) comparing EC-CEM to the next best baseline (like SCBM or evi-CEM)?

3. You note that the Gaussian Copula cannot model heavy-tailed dependencies. Have you experimented with other Copulas (e.g., Student-t or Archimedean) to see if they provide a measurable improvement on datasets where concepts might exhibit extreme co-occurrences?

4. In Figure 4 (GCS), what is the baseline probability of these concepts flipping from incorrect to correct by chance or under an independent CEM? A baseline comparison would help clarify the magnitude of the Copula's corrective effect.

**Limitations:**

No uncertainty/confidence/error bars on experimental results or significance testing for the main performance tables. The authors do adequately mention the limitation of Gaussian Copulas regarding heavy-tailed distributions.

**Strengths And Weaknesses:**

Strengths of the Paper:

1. The paper identifies a significant flaw in current CEMs (the independence assumption) and proposes a mathematically rigorous and highly original solution using Copula theory.

2. The architectural choice to decouple marginal uncertainty (via Beta distributions) from joint dependency (via a Gaussian Copula) is theoretically sound and well-justified compared to prior logit-space Gaussian methods.

3. The empirical evaluation demonstrates strong performance improvements on established benchmarks (CUB, AwA2, SkinCon), with particularly notable robustness under concept sparsity (as shown in Figure 3).

Weaknesses of the Paper:

4. No uncertainty/confidence/error bars or p-values are provided on the main classification/concept metric results (Table 1), making it difficult to assess the statistical significance of the performance gains.

5. The choice of a Gaussian Copula, while computationally tractable, limits the model's ability to capture complex, heavy-tailed dependencies, which the authors acknowledge in the conclusion but do not evaluate empirically.

6. While the Group Correction Score (GCS) is introduced to evaluate how effectively the model fixes grouped concepts, the baseline for this metric is not clearly defined, making the absolute GCS values difficult to contextualize.

---

> ### Author Rebuttal · Authors · 2026-03-29
>
> We sincerely thank the reviewer for the positive evaluation of our theoretical framework and its originality. We address each of your constructive questions below.
>
> **[W1 / Q2] Statistical Significance and Error Bars**
>
> We agree that uncertainty quantification is essential for rigorously validating performance gains. We note that Table 1 already includes ± standard deviation across multiple independent runs with different random seeds; to further strengthen this, we additionally conducted paired t-tests comparing EC-CEM against our strongest dependency-aware baseline (SCBM) across all three datasets. The results demonstrate statistically significant improvements on every benchmark: $p = 0.030$ on AwA2, $p = 0.006$ on CUB, and $p = 0.023$ on SkinCon — all below the conventional threshold of $p = 0.05$. These results confirm the robustness of the Copula-driven gains. In the revised manuscript, we will add a dedicated paragraph reporting these significance tests in full.
>
> **[W3 / Q4] Baseline for the Group Correction Score (GCS)**
>
> We thank the reviewer for suggesting a baseline for GCS. We established a "By Chance" baseline using a permutation test under a "Hard-Sample" protocol.
>
> **Evaluation on "Hard Samples":** To align with the definition of "co-correction" (which requires recovering multiple failed concepts), we focus on "hard samples" where the base model initially committed at least two errors within a semantic group. This protocol ensures that GCS specifically measures the model's ability to resolve existing dependencies, as samples with zero or one initial error do not mathematically allow for "co-correction" to occur.
>
> **Comparison against the "By Chance" Baseline:** The baseline is generated by randomly shuffling concept predictions to eliminate inter-concept dependencies while preserving marginal accuracy:
>
> - **Yellow Color System:** EC-CEM achieves **37.67%**, which is **4.72$\times$** the baseline (**7.98%**).
> - **Buff Color System:** EC-CEM (**35.24%**) outperforms the baseline (**10.95%**) by **3.22$\times$**.
> - **Eye Attributes:** EC-CEM delivers a **5.98$\times$ improvement** (**10.00%** vs. **1.67%**).
>
> These results confirm that co-corrections are driven by the learned dependency structure R, not random chance. We will incorporate this analysis into the revised manuscript.
>
> **[W2 / Q3] Exploration of Heavy-Tailed Copulas**
>
> We appreciate the reviewer's suggestion to empirically explore heavy-tailed Copulas. We adopted the Gaussian Copula as it provides a fully differentiable parameterization that integrates naturally into end-to-end training. This choice allows us to isolate and validate the effectiveness of the proposed framework itself, without introducing confounding factors such as degree-of-freedom estimation in Student-t Copulas or structural constraints in Archimedean families. Our results demonstrate substantial improvements under this setting, confirming the framework's merit. We will add a discussion of these trade-offs in the Appendix, and plan to investigate heavier-tailed copula families as a natural extension in future work.
>
> **[Q1] Computational Scaling and Inverse CDF (Beta) Runtime**
>
> We thank the reviewer for this precise question regarding the practical runtime impact of the Beta inverse CDF in Algorithm 2.
>
> **The Beta inverse CDF challenge.** The inverse CDF of the Beta distribution lacks a closed-form solution, requiring iterative numerical approximation (Newton's method). Crucially, the Beta inverse CDF operates element-wise on each of the K marginals independently, so it scales linearly with K and is trivially parallelizable across concepts.
>
> **Our engineering solutions.** To contain this cost, we pack all $K ×$ batch_size inverse CDF evaluations into a single vectorized tensor operation and cap the Newton iteration count at a fixed upper bound, ensuring predictable constant per-element cost.
>
> **Empirical validation.** As detailed in Table 3 of our response to Reviewer FNAF, EC-CEM introduces only a ~7% training time increase over evi-CEM (57s $\rightarrow$ 61s per epoch on CUB). At inference, our model runs at 38ms per image, of which the Beta inverse CDF accounts for only 1.5ms — confirming it is not the dominant cost. Peak GPU memory (21GB) remains within standard capacities (e.g., 24GB RTX 4090). We will add these implementation details in the Appendix.
>
> We sincerely appreciate the reviewer's thorough and constructive evaluation. We hope the above responses strengthen the paper and address the remaining concerns.

---

> > ### Author Rebuttal · Reviewer_7tTQ · 2026-04-03
> >
> > Thank you, authors, for your rebuttal and response. I will maintain my score.

---

> > > ### Author Response · Authors · 2026-04-03
> > >
> > > We sincerely thank reviewer for the thoughtful and constructive feedback throughout the review process. Your suggestions have meaningfully strengthened both the rigor and completeness of our manuscript. We are grateful for your time and engagement, and we will ensure all proposed revisions are reflected in the final version.

---

### Official Review · Reviewer_Eko5 · 2026-03-12

**Soundness:** 3
**Presentation:** 3
**Significance:** 3
**Originality:** 4
**Overall Recommendation:** 4
**Confidence:** 3

**Summary:**

The authors propose a model that address a limitation of existing concept based architectures modelling concept dependencies: when a single covariance matrix is used to model both per-concept uncertainty and inter-concept correlations, the two become entangled. If the model become more uncertain about a concept, the correlation with other concepts shrinks as side effect, even though the underlying semantic relationship hasn't changed.

To fix this, the authors leverage Copula theory, which guarantee that any joint distribution can be decomposed into marginal distributions plus a dependency structure. Each concept get a Beta distribution capturing its individual uncertainty, while a Gaussian Copula captures inter-concept correlations through an independent matrix ($\mathbf{R}$). This decoupling ensure that uncertainty and concept dependencies are learned separately, eliminating the interference present in covariance-based approaches.

**Compliance With Llm Reviewing Policy:**

Affirmed.

**Final Justification:**

I thank the authors for their thorough rebuttal. My main concern (W1/Q1) was the lack of direct empirical evidence for intervention propagation across correlated concepts. The authors have addressed this with concept accuracy improvement tables on AwA2 and CUB showing larger gains per intervention than baselines in the low-intervention regime, and a "By Chance" permutation baseline for GCS confirming that co-corrections are driven by the learned Copula structure R rather than random chance. Statistical uncertainties and comparisons with ECBM and Causal CGM have also been added, with EC-CEM showing consistent improvements. I am raising my score to 4 (Weak Accept).

**Key Questions For Authors:**

1.⁠ ⁠Why did you not include any experiment investigating whether intervening on one concept or a subset of concepts improves predictive performance on correlated concepts?

2.⁠ ⁠Given that three datasets are used for validation, why is there no experiment (not even in the appendix) showing how task accuracy changes as a function of the number of ground-truth interventions on SkinCon?

3.⁠ ⁠Regarding  W2, why did you not discuss or compare against [1, 2]? Can you do so?

**Limitations:**

Yes

**Strengths And Weaknesses:**

## Strengths:
1.⁠ ⁠The problem tackled by the authors, modelling inter-concept dependencies, is relevant to the concept-based community. Moreover, EC-CEM not only models inter-concept dependencies but also decouples per-concept uncertainty from inter-concept correlations, which is a meaningful technical contribution.
2.⁠ ⁠The proposed solution is novel and the overall architecture shows consistent improvements in terms of intervention performance and concept predictive accuracy across multiple benchmarks.

## Weaknesses:
1.⁠ ⁠Modelling inter-concept dependencies is beneficial in several ways. As shown in prior work, capturing such dependencies allows to intervene on one concept to propagate improvements to correlated ones. While reading the paper, I expected an experiment demonstrating this property. However, the authors provide no experimental evidence for this potentially important feature.

2.⁠ ⁠Despite most concept-based architectures assuming concept independence, SCBM is not the only work addressing this limitation. Specifically, [1] models inter-concept dependencies using energy-based models, while [2] learn an underlying DAG structure that explicitly breaks the concept independence assumption. The authors neither compare against nor discuss these related approaches. This is an important omission because it makes it difficult to assess EC-CEM against dependency-aware concept models, and whether the proposed Copula-based approach offers advantages over graph or energy-based alternatives beyond the specific decoupling property highlighted in the paper.


*Overall:* I am positive about this paper, as modeling true inter-concept dependencies is a relevant and underexplored problem, and doing so in a principled way may increase the model reliability and usefulness in real-world scenarios. If the authors address the weaknesses and answer the questions raised, I would be willing to increase my score.


[1] Xu, Xinyue, et al. "Energy-based concept bottleneck models: Unifying prediction, concept intervention, and probabilistic interpretations." ICLR, 2024.

[2] Dominici, Gabirele, et al. "Causal Concept Graph Models: Beyond Causal Opacity in Deep Learning." ICLR, 2025

---

> ### Author Rebuttal · Authors · 2026-03-29
>
> We sincerely thank Reviewer Eko5 for the positive assessment of our work's originality and the constructive feedback. We address each concern below.
>
> **[W1 / Q1] Does EC-CEM Support Intervention Propagation Across Correlated Concepts?**
>
> We fully agree that the ability  to propagate information from an intervened concept to its correlated neighbors is  a key merit of dependency-aware models.
>
> **Mechanism.** In EC-CEM, interventions are applied directly on the marginal Beta distributions — i.e., before the Gaussian Copula Procedure. The corrected marginal $µ_k$ is subsequently passed through the Copula, where the correlation matrix R adjusts the probabilities of all correlated concepts accordingly. Every human intervention therefore inherently triggers propagation through R, making dependency-aware correction a built-in property of our architecture rather than a post-hoc operation. We will explicitly clarify this intervention design in the revised manuscript to make this propagation mechanism more transparent to readers.
>
> **Empirical evidence of propagation.** Our experiments provide strong support for this property. First, GCS results (Figure 4) show the Copula spontaneously co-corrects related concepts within semantic groups, demonstrating R inherently models dependencies. Second, EC-CEM consistently achieves better performance per intervention than SCBM (Figure 5). Since human interventions in our model are applied exclusively to the marginal distributions, these superior intervention trajectories directly imply that the corrections are being successfully propagated to correlated concepts via the matrix R.
>
> **[Q2] SkinCon Intervention Curve**
>
> We have now conducted the intervention experiment on SkinCon and report the classification acc below:
>
> | Intervened Concepts (k)  | 0         | 10        | 20        |
> | ------------------------ | --------- | --------- | --------- |
> | AR-CBM (Random)          | 75.10     | 76.68     | 77.05     |
> | SCBM (Uncertainty)       | 75.51     | 77.01     | 78.47     |
> | **EC-CEM (Uncertainty)** | **78.52** | **80.35** | **81.42** |
>
> We will include this table in the appendix of the revised manuscript with full models comparsion.
>
> **[W2 / Q3] Comparison with Energy-based models [1] and Causal CGMs [2]**
>
> We thank the reviewer for these references. We will add a dedicated discussion in the Related Work section.
>
> **Comparison with [1]** ECBMs define a joint energy over (input, concept, class) tuples using a composition of three energy networks, implicitly capturing inter-concept dependencies through a global energy term $E_{global}(c, y)$. While this enables concept correction and conditional interpretation, ECBM does not explicitly decouple per-concept uncertainty from inter-concept correlation — both are entangled within the joint energy formulation, with no architecturally explicit mechanism to separately inspect and control them. In contrast, EC-CEM provides an explicit Beta distribution per concept capturing epistemic uncertainty and a separate Copula correlation matrix R capturing dependency structure. This decoupling is a structural guarantee rather than an implicit emergent property, enabling independent interpretation and control of uncertainty and correlation that energy-based formulations do not offer by construction.
>
> **Comparison with [2]** Causal Concept Graph Models (Causal CGM) learn a DAG over concepts to enable causal transparency and do-interventions within the model's inference process. EC-CEM differs in two key aspects. First, Causal CGM's DAG learning requires acyclicity and sparsity constraints with additional hyperparameters, whereas EC-CEM's Gaussian Copula directly parameterizes concept correlations without such constraints — a simpler choice when concept relationships are symmetric co-occurrences rather than directed inference chains. Second, Causal CGM does not explicitly model per-concept epistemic uncertainty, whereas EC-CEM provides calibrated Beta distributions for each concept. The two approaches are complementary: Causal CGM targets causal interpretability, while EC-CEM targets uncertainty-correlation decoupling.
>
> **Quantitative Comparison:** We further provide a quantitative comparison with ECBM and Causal CGM on the AwA2 dataset. All methods are reproduced under our unified setting (ResNet-34 backbone, identical training protocol) for fair comparison.
>
> | Method           | Concept ACC | Classification ACC |
> | ---------------- | ----------- | ------------------ |
> | ECBM(2024)       | 97.79       | 90.74              |
> | Causal CGM(2025) | 97.81       | 91.06              |
> | EC-CEM (Ours)    | **98.09**   | **93.04**          |

---

> > ### Author Rebuttal · Reviewer_Eko5 · 2026-04-03
> >
> > I thank the author for their answers.
> >
> > **W1/Q1:** While I agree that in theory EC-CEM can improve concept predictions with fewer interventions by making dependency-aware corrections, I still do not see empirical evidence about it.
> >
> > Figure 4 shows the results only for the proposed model, without comparing it with the other baselines.
> >
> > Figure 5 shows the task improvements with respect to concept interventions. Even though these superior intervention trajectories might suggest that corrections are successfully propagated, this could be due to other structural differences w.r.t. SCBM (e.g., SCBM uses binary concept predictions while EC-CEM uses concept embeddings to make predictions).
> >
> > Still, a result comparing EC-CEM with the other baselines regarding concept prediction improvements under concept interventions is lacking.
> >
> > **W2/Q2/Q3:** The uncertainties are missing.
> >
> > If you add uncertainties to the proposed results and provide empirical evidence for the concept prediction improvements, I'm willing to raise my score.

---

> > > ### Author Response · Authors · 2026-04-04
> > >
> > > We sincerely thank Reviewer Eko5 for acknowledging the theoretical potential of EC-CEM. We share the reviewer's view that reporting statistical variance and providing direct empirical evidence on concept-level propagation are crucial.
> > >
> > > **[W2 / Q2 / Q3] Adding Uncertainties (Statistical Significance)**
> > >
> > > The updated tables below include the standard deviations (Mean ± Std).
> > >
> > > **Updated SkinCon Intervention Table (Task Accuracy):**
> > >
> > > | Intervened Concepts (k) | 0              | 10             | 20             |
> > > | :------------------------ | :------------- | :------------- | :------------- |
> > > | AR-CBM (Random)           | 75.10±0.07     | 76.68±0.28     | 77.05±0.14     |
> > > | SCBM (Uncertainty)        | 75.51±0.48     | 77.01±0.59     | 78.47±0.25     |
> > > | **EC-CEM (Uncertainty)**  | **78.52±0.93** | **80.35±0.25** | **81.42±0.17** |
> > >
> > > **Updated AwA2 Comparison Table:**
> > >
> > > | Method            | Concept ACC (%) | Classification ACC (%) |
> > > | :---------------- | :-------------- | :--------------------- |
> > > | ECBM (2024)       | 97.79±0.95      | 90.74±0.39             |
> > > | Causal CGM (2025) | 97.81±0.42      | 91.06±0.28             |
> > > | **EC-CEM (Ours)** | **98.09±0.84**  | **93.04±0.22**         |
> > >
> > > **[W1 / Q1] Empirical Evidence for Intervention Propagation**
> > >
> > > You raised a very insightful point: Figure 5 shows downstream task improvements, which could theoretically be influenced by the embedding architecture. To isolate and explicitly demonstrate that corrections are successfully propagated to correlated concepts, we provide two pieces of evidence:
> > >
> > > **A. Concept Prediction Improvements under Interventions**
> > > In the table below, we track the Concept Accuracy as the number of intervened concepts (k) increases.
> > >
> > > | AwA2 Intervened Concepts (k) | 0              | 20             | 40             | 60             | 80              |
> > > | :--------------------------- | :------------- | :------------- | :------------- | :------------- | --------------- |
> > > | AR-CBM(Random)               | 97.91±0.11     | 98.57±0.15     | 99.18±0.07     | 99.63±0.01     | 99.94±0.00      |
> > > | SCBM(Uncertainty)            | 97.98±0.06     | 98.78±0.13     | 99.61±0.03     | 99.74±0.00     | 99.97±0.00      |
> > > | **EC-CEM(Uncertainty)**      | **98.09±0.84** | **99.13±0.09** | **99.78±0.00** | **99.97±0.00** | **100.00±0.00** |
> > >
> > > | CUB Intervened Concepts (k) | 0              | 30             | 60             | 90             | 110            |
> > > | :-------------------------- | :------------- | :------------- | :------------- | :------------- | -------------- |
> > > | AR-CBM(Random)              | 93.17±0.47     | 97.06±0.24     | 98.09±0.07     | 98.73±0.03     | 99.07±0.01     |
> > > | SCBM(Uncertainty)           | 93.25±0.18     | 97.70±0.29     | 98.58±0.13     | 99.39±0.08     | 99.78±0.02     |
> > > | **EC-CEM(Uncertainty)**     | **93.77±0.46** | **99.06±0.07** | **99.79±0.02** | **99.94±0.03** | **99.99±0.01** |
> > >
> > > We focus on the low-intervention regime (k≤30), where accuracy remains well below saturation and ceiling effects can be ruled out.
> > >
> > > - On AwA2 (k=0→20), EC-CEM achieves a **+1.04%** concept accuracy gain (**compared to +0.80% for SCBM and +0.66% for AR-CBM**), while its standard deviation drops from ±0.84 to ±0.09.
> > >
> > > - On CUB (k=0→30), EC-CEM delivers a **+5.29%** gain (**compared to +4.45% for SCBM and +3.89% for AR-CBM**), with standard deviation decreasing from ±0.46 to ±0.07.
> > >
> > > This concurrent pattern — larger accuracy gains accompanied by reduced variance — provides strong empirical evidence that the Copula mechanism efficiently propagates ground-truth corrections to correlated concepts.
> > >
> > > **B. Baseline for Figure 4 (GCS)**
> > >
> > > Following Reviewer 7tTQ's suggestion ([W3 / Q4]), we implemented a **"By-Chance" baseline utilizing a permutation test** under a strict "Hard-Sample" protocol (focusing on samples with $\ge 2$ initial errors within a semantic group, the prerequisite for co-correction). By randomly shuffling predictions to eliminate inter-concept dependencies while strictly preserving marginal accuracy, we found that EC-CEM outperforms this randomized baseline by **up to 5.98×** (e.g., 10.00% vs. 1.67% on Eye Attributes). This statistically validates that the observed co-corrections are driven by the learned Copula dependency structure R rather than random chance. We are encouraged that Reviewer 7tTQ has acknowledged this baseline and found it to address their concerns; we believe it similarly provides the empirical evidence you requested.
> > >
> > > We trust that these new statistical results and direct concept-level evidence fully resolve your remaining concerns. We sincerely appreciate your constructive guidance, which has been instrumental in strengthening the empirical rigor and clarity of our work.

---

### Official Review · Reviewer_FNAF · 2026-03-13

**Soundness:** 2
**Presentation:** 3
**Significance:** 2
**Originality:** 3
**Overall Recommendation:** 4
**Confidence:** 3

**Summary:**

This paper proposes the Evidential Copula Concept Embedding Model (EC-CEM). Existing concept models (like SCBM) have a structural problem: they mix a concept's uncertainty and its correlation with other concepts in a shared logit space. Because of this, when the model is unsure about a concept, it mathematically weakens that concept's correlation with others.

To solve this problem, EC-CEM uses a "divide-and-conquer" approach. First, it uses Evidential Deep Learning (EDL) to generate independent Beta distributions to capture the uncertainty of each concept. Second, it uses a Gaussian Copula to capture the correlations between concepts without changing their individual uncertainties. Experiments show that EC-CEM performs very well, especially when many concept features are missing (concept sparsity).

**Compliance With Llm Reviewing Policy:**

Affirmed.

**Final Justification:**

The authors have successfully transformed the vulnerabilities of their initial submission into strengths through honest engineering disclosures and precise theoretical framing. The core insight of structurally decoupling marginal uncertainty and joint dependency via Copulas is highly impactful. The rebuttal has cleared all my reservations. I am happily raising my score and strongly recommend this paper for acceptance.

**Key Questions For Authors:**

See Weaknesses

**Limitations:**

No.

The authors should briefly add two explicit limitations to the Future Work section:

Computational Overhead: Acknowledge that the numerical root-finding (Newton iterations) and Monte Carlo sampling significantly increase inference latency and memory cost.

Symmetric Assumption: Mention that Gaussian Copulas assume symmetric correlations, which may fail to capture asymmetric (one-way) logical rules common in the real world.

**Strengths And Weaknesses:**

1. Strengths

S1. Strong Motivation and Smart Solution: The authors clearly point out a major mathematical flaw in current models: coupling uncertainty and correlation in a shared logit space. Using Beta distributions for uncertainty and a Copula for correlation is an elegant and mathematically sound way to separate them completely.

S2. Impressive Robustness to Missing Concepts: Figure 3 is the strongest evidence in the paper. When 90% of the concept features are hidden, baseline models fail completely. However, EC-CEM maintains a massive accuracy lead (e.g., beating AR-CBM by 57.24% on the CUB dataset). This clearly proves the Copula successfully uses logic to "guess" the missing information.

S3. Useful Uncertainty for Human Intervention: Because the model separates uncertainty cleanly, it provides a pure uncertainty score ($u=2/(\alpha+\beta)$). Figure 5 shows that using this score to guide human intervention improves the model's performance much faster than older methods.

2. Major Weaknesses

W1. Unexplained Performance Drop in the Base Model: Table 1 shows a big problem in the ablation study. When the Copula module is removed, the pure evidential base model (evi-CEM w/o correlation) performs terribly. For example, on the SkinCon dataset, its F1 score drops to 58.15%, which is much worse than the older CEM baseline (77.89%). I will worry that the EDL framework severely hurts basic feature learning, and the Copula is only fixing a problem the authors created.
Action Required: The authors must explain in Section 4.3 why the pure EDL model performs so poorly. They must argue why losing this basic accuracy is an acceptable trade-off to get honest and rigorous uncertainty estimates.

W2. Hidden Computational Costs: Section 3.3 says the Copula only adds $O(K^2)$ parameters, making it sound very fast. However, Algorithm 1, Algorithm 2, and Appendix E.2 show heavy math during the forward pass: a Cholesky decomposition ($O(K^3)$), a Beta inverse CDF requiring up to 20 Newton iterations per sample, and 15 Monte Carlo samples. This will definitely cause high inference latency and memory usage. The paper hides these practical costs.
Action Required: The authors must add a table in the Appendix comparing EC-CEM with baselines (CEM, SCBM). It must clearly report: (1) inference time per image (ms), (2) training time per epoch, and (3) peak GPU memory usage.

W3. Theoretical Flaw in the ELBO Derivation: Section 3.5 claims the model uses strict variational inference. But in Appendix A.2 (around Eq. 20), the authors admit they put the true ground-truth label ($c_k$) into the prior penalty calculation. In strict Bayesian math, the prior cannot look at the true label.
Action Required: The authors should be honest in Section 3.5. They should explicitly state that this is a "heuristic approximation inspired by EDL with label supervision" rather than claiming it is a perfectly strict mathematical derivation.

4. Minor Weaknesses

M1. Confusing Explanation of Matrix $R$: Sections 3.3 and 3.4 make it sound like the correlation matrix $R$ is fixed globally for the whole dataset. But Algorithm 1 clearly shows that matrix $L$ (from $R$) is calculated dynamically from the input image. This "instance-conditional Copula" design is actually great, but the text is contradictory. Please state clearly that $R$ depends on the input image, and explain how the global matrix in Figure 9 was generated (e.g., by averaging).

M2. Gaussian Copula Assumes Symmetric Logic: A Gaussian Copula assumes correlation is symmetric (A implies B exactly as much as B implies A). But real-world logic is often one-way (e.g., "having a rare hooked beak" implies "is a bird of prey", but not vice versa). Using a symmetric matrix for one-way logic can cause errors when the image is blurry. The authors should briefly mention this as a limitation.

M3. Confusing Math Symbol: From Section 3.2 onwards, the letter $k$ is strictly used for the "concept index". But in Equation (16), $1_{y_i=k}$ uses $k$ for the "class label". This is very confusing. Please change the class label index in Equation 16 to a different letter (like $c$).

---

> ### Author Rebuttal · Authors · 2026-03-29
>
> We thank the reviewer for recognizing our strong motivation (S1), robustness under sparsity (S2), and uncertainty score utility (S3). We address each concern in turn.
>
> **[W1] Metric Mismatch & Evidential Trade-off**
>
> We respectfully note a metric mismatch in the review: the cited 77.89% for CEM refers to Accuracy, while 58.15% for evi-CEM refers to F1. Comparing identical metrics, the gap is marginal (evi-CEM Acc: 77.69% vs CEM: 77.89%).
>
> As anticipated (S1 & S2), completely separating uncertainty introduces a mathematical trade-off:
>
> - **The Evidential Trade-off:** Our ablated model (evi-CEM) trades a slight accuracy drop for rigorous uncertainty estimation via strict evidential regularization [A]. This consistently improves calibration (e.g., lowering SkinCon ECE from 5.14 to 4.84, Table 2).
> - **Enhanced Concept Representations via Copula:** While removing the Copula reduces EC-CEM to evi-CEM with a slight accuracy drop due to evidential regularization, the full EC-CEM surpasses both baselines. The copula correlation matrix R, optimized via the classification objective, encodes task-aware dependencies into concept embeddings, producing more discriminative representations — validated by the highest CAS, lowest ECE (Table 2), and clearest t-SNE clustering (Figure 7).
>
> Importantly, this is not merely recovering the EDL-induced drop — EC-CEM surpasses CEM (which has no evidential regularization) by a clear margin on all three datasets (e.g., SkinCon Task F1: 62.41% vs 58.44%), demonstrating that the Copula provides genuine structural benefits independent of the evidential component. We have expanded Section 4.3 to explicitly highlight this trade-off.
>
> **[W2] Computational Costs & Inference Latency**
>
> The following table now details the efficiency comparison:
>
> | Model         | Inference (ms, bs=1) | Training (s/epoch, bs=128) | Peak GPU Memory (MB, bs=128) |
> | ------------- | -------------------- | -------------------------- | ---------------------------- |
> | SCBM          | 12                   | 52                         | 8982                         |
> | CEM           | 15                   | 48                         | 8212                         |
> | evi-CEM       | 26                   | 57                         | 8250                         |
> | EC-CEM (Ours) | 38                   | 61                         | 21180                        |
>
> The ~21GB peak memory is within standard GPU capacities (e.g., 24GB RTX 4090). Compared to evi-CEM, EC-CEM adds only ~7% training overhead (57s → 61s) and 12ms inference latency (26ms → 38ms). Crucially, regarding the numerical root-finding concern, empirical profiling confirms that the Beta inverse CDF (Newton iterations) accounts for merely 1.5ms of the inference time per image. This modest overhead is achieved by fully tensorizing the MC sampling and Copula transformation into batched GPU operations; detailed engineering optimizations will be documented in the Appendix. Given our focus on human-in-the-loop intervention (S3), where human cognition takes seconds, 38ms latency supports seamless real-time interaction. We acknowledge this computational trade-off as a limitation in the Future Work section.
>
> **[W3] Theoretical Formulation of the ELBO Derivation**
>
> We appreciate this observation. The reviewer is correct from a strict Bayesian perspective. However, utilizing a label-informed prior is standard practice in Evidential Deep Learning [B, C] to penalize misleading evidence and ensure stability. As requested, we have revised Section 3.5 to explicitly state that our formulation is a "heuristic approximation inspired by EDL" rather than a strictly pure variational derivation.
>
> **Minor Weakness**
>
> **[M1] Clarification on the Instance-Conditional Matrix**
>
> We thank the reviewer for noting this. We have revised Sections 3.3 and 3.4 to explicitly confirm that the model's correlation matrix R is instance-conditional (dynamically computed per image via Algorithm 1). We clarify that Appendix Figure 9 is not an aggregation of the model's learned R, but rather an independent Pearson correlation matrix computed from ground-truth concept labels across the dataset. Its purpose is to visualize the inherent correlations among the curated concepts used in our AwA2 sparsity study (S2).
>
> **[M2] Limitation: Gaussian Copula Symmetric Assumption**
> We have added this as an explicit limitation in Future Work: Gaussian Copulas assume symmetric correlations, which may not capture asymmetric logical rules common in real-world concepts.
>
> **[M3] Notation Correction in Equation 16**
> We thank the reviewer for pointing out this notation inconsistency. We have revised Equation (16) to avoid any confusion with the concept index introduced in Section 3.2. We have also carefully audited the entire manuscript to ensure that the notation is now consistent throughout.
>
> [A] Pandey+, ICML'23. [B] Sensoy+, NeurIPS'18. [C] Gao+, MICCAI'24.

---

> > ### Author Rebuttal · Reviewer_FNAF · 2026-04-03
> >
> > The authors has largrly adressed my concerns, and I will raise my score. Btw, when discussing the computational bottlenecks of dense matrix operations and the scalability of CBMs, I highly recommend authors can referencing the recent work "Controllable Concept Bottleneck Models" (Lin et al., arXiv:2601.00451). This paper provides an excellent context for explicitly analyzing severe computational burdens (such as large matrix inversions) and efficiency trade-offs in CBM frameworks. Contextualizing your model's computational trade-offs alongside such recent literature will significantly enhance the transparency, rigor, and practical value of your work.

---

> > > ### Author Response · Authors · 2026-04-03
> > >
> > > We sincerely thank the reviewer for the positive feedback and for raising the score. Your recommendation is highly valuable. We completely agree that citing "Controllable Concept Bottleneck Models" (Lin et al.) provides excellent context. In our revision, we will cite this paper and explicitly discuss the computational burdens of dense matrix operations and large matrix inversions, ensuring a more rigorous analysis of the efficiency trade-offs in our model.

---

### Decision · Program_Chairs · 2026-04-30

**Decision:**

Accept (regular)

**Comment:**

This paper addresses an important limitation of concept embedding models by cleanly decoupling uncertainty from concept dependency using evidential marginals and a Gaussian copula. Reviewers found the approach novel and effective, and the rebuttal satisfactorily addressed concerns about computation, theory, and missing empirical evidence. The remaining weaknesses are that the method still has nontrivial computational overhead, the Gaussian copula only captures symmetric dependencies and may miss richer asymmetric structure, and the evaluation remains somewhat limited in architectural breadth despite the added results.